# The genetic landscape of human functional brain connectivity

**Bernardo de APC Maciel** [1], **Marijn Schipper** [1], **Cato Romero**[2], **Christiaan de Leeuw** [1], **Koen Helwegen**[1], **Danielle Posthuma**[1,3], **Jeanne E. Savage** [1] **& Martijn P. van den Heuvel** [1,2] ✉

Investigating the genetic underpinnings of functional brain connectivity is essential to understand how genetic variation influences brain health and disease. Here, a mass-univariate approach was adopted to study the genetic architecture of functional brain circuitry ($N_{total}$ = 28,159 subjects) with high spatial resolution (82 brain regions). Common genetic variants explained individual differences in 33% of all 3321 inter-regional functional pathways with 72 significant associations reflecting widespread, pleiotropic effects across the connectome. These associations were mapped to five genes—*PAX8, EphA3, SLC39A12, THBS1* and *APOE*—with known associations with brain phenotypes and which converged in biological processes related to neurodevelopment and cardiovascular and cognitive traits (enrichment minimum $p = 3.0 \times 10^{-6}$ and $p = 1.6 \times 10^{-5}$, respectively). Our findings show that the genetic component of individual differences in functional brain connectivity is largely shared throughout the brain, highlighting the importance of genetic variation in large-scale brain organisation and its relationship with cognitive function and overall health.

Synchronisation in neural activity between regions is observed throughout the human brain[1]. These macroscale dynamic activity patterns form distinct functional networks of activity across multiple neural systems and are important for healthy brain functioning, cognition and behaviour[2–4]. Twin and family studies show that these functional networks are heritable[5–7]. Furthermore, recent large-scale Genome-Wide Association Studies (GWAS) have identified relevant genetic loci underlying this heritability[8–12].

Several genes play an important role in macroscale brain properties. Activity of the default mode network–a central brain network involved in cognition, self-awareness and theory of mind–is associated with the *APOE* gene, a well-known genetic factor associated with longevity, brain atrophy and Alzheimer's disease[8,13,14]. Genes *GPR139, DACH1* and *APOE* are associated with trajectories of lifetime

trajectories of brain atrophy and partake in metabolic processes relevant to both early brain development and neurodegeneration[8]. *TOMM40, APOE* and *APOC1* are associated with functional connectivity (FC) of brain networks, showing that genetic factors may have widespread effects throughout the brain[10]. Moreover, genetic associations are often shared across neuropsychiatric[15–17] and magnetic resonance imaging (MRI)-derived[17–19] traits, pointing towards common biological processes driving changes in these phenotypes and suggesting that genetic differences might underlie heterogeneity (i.e. individual differences) in FC. While previous studies have identified loci associated with specific functional circuitry[20], resting state networks[8], independent FC components[10], and functional graph metrics[21,22], these often have limited spatial resolution, limiting the understanding of the specific regional links that may be critical for brain health. Identifying

[1]Department of Complex Trait Genetics, Center for Neurogenomics and Cognitive Research, Amsterdam Neuroscience, Vrije Universiteit Amsterdam, Amsterdam, the Netherlands. [2]Department of Child and Adolescent Psychiatry and Psychology, Section Complex Trait Genetics, Amsterdam Neuroscience, Vrije Universiteit Medical Center, Amsterdam UMC, Amsterdam, the Netherlands. [3]Department of Clinical Genetics, Section Complex Trait Genetics, Amsterdam Neuroscience, Vrije Universiteit Medical Center, Amsterdam University Medical Centre, Amsterdam, the Netherlands. ✉e-mail: martijn.vanden.heuvel@vu.nl

genetic variants that influence a large share of the functional connectome and the specific regions associated with different biological processes is crucial for understanding the core biological pathways that shape large-scale brain network organisation and its disruption in disease.

In this study, we characterise the global genetic architecture of human functional brain connectivity. We perform mass-univariate GWAS of 3321 pairwise connections between 82 brain regions measured through MRI in a sample of 28,159 individuals of European ancestry in the UK Biobank cohort[23]. We identify and describe genetic associations with individual functional connections in the brain. Then, we compare the genetic signal across different functional brain connections and quantify the extent of genetic effects throughout the brain. Specific locus–edge and gene–edge associations are mapped using gene-set and pathway enrichment analysis to explore putative biological processes underlying heterogeneity in brain circuitry. Altogether, this study identifies common genetic variants and biological pathways associated with functional brain connectivity, providing mechanistic insights into how genetic variation may influence individual differences in functional connectivity.

## Results

### Common genetic variants explain individual differences in the strength of functional connections in the human brain

We conducted 3321 GWAS on functional connectivity between all brain connections across the human brain in a discovery sample of 24,442 subjects. These comprise pairwise functional connections between 82 brain regions: 68 anatomically-defined cortical areas and 14 subcortical nuclei (Methods; Supplementary Note 1; Overview of methods and results in Supplementary Fig. 1).

Single Nucleotide Polymorphism (SNP)-based heritability ($h^2_{SNP}$) was calculated with Linkage Disequilibrium Score Regression (LDSC; Methods)[24] to estimate the global influence of common genetic variants in all functional brain connections (minor allele frequency > 1%). The median estimated $h^2_{SNP}$ across all 3321 edges was 3.63% and nominally significant effects were found in 1083 edges (32.6% of total) with a median $h^2_{SNP}$ of 6.2%. The most heritable edge ($h^2_{SNP} = 19.9\%$, SE = 2.5%) connected left supramarginal gyrus and left middle temporal gyrus, two brain regions involved in language processing (Supplementary Data 1; Fig. 1a)[25–27]. Median phenotypic correlation between edges was 0.002 (range = [−0.681; 0.953]). Heritability estimates of subcortical edges were lower than average (mean subcortical $h^2_{SNP} = 1.9\%$), likely due to the relatively low signal-to-noise ratio of FC between subcortical nuclei[27]. These results indicate that a moderate amount of interindividual differences in FC can be explained by interindividual differences in common genetic variants.

Subnetwork enrichment analysis was performed to determine if brain connections within specific resting-state networks (RSNs) were under stronger genetic control than those in other or between RSNs. To this end, we aggregated the results for the individual edges into RSNs. RSNs were defined by assigning cortical edges to one of the seven Yeo-Krienen RSNs or to the subcortical network if both regions connected by an edge belonged to the same RSN (Supplementary Methods). To account for the higher correlation of edges within the same network, a linear model was developed to aggregate the $h^2_{SNP}$ of individual edges and calculate RSN enrichment for $h^2_{SNP}$ (Methods). Out of all RSNs (median $h^2_{SNP}$ range = [1.9; 6.9]%), the Ventral Attention network was the most enriched for $h^2_{SNP}$ (median $h^2_{SNP} = 6.9\%$; $t(269) = 2.27$, $p = 0.01$; Table 1; Fig. 1b), but did not survive multiple-testing correction.

### Seventy-eight replicated locus–edge associations within the human functional connectome

Genomic loci associated with FC were identified in the discovery sample of 24,451 subjects and replicated in 3708 subjects across 3321

functional edges ("Methods"). Across all GWAS, a total of 208 significant locus–edge associations were identified in the discovery sample (two-sided $\alpha_{disc} = 5 \times 10^{-8}/3321$ for study-wide significance (SWS); Methods). Of those, 78 univariate locus–edge associations involving 27 distinct edges were replicated in the holdout sample (two-sided $\alpha_{rep} = 0.05/208$; Supplementary Data 2).

The 78 SWS loci were often overlapping across edges or in close proximity in the genome. Locus–edge associations overlapping in the same genomic region were found to be colocalised in the discovery sample. This indicates that the same putative causal variants in these loci contribute to interindividual variability in multiple edges across the connectome (Supplementary Note 4; Supplementary Data 3). These locus–edge associations were aggregated into 4 non-overlapping genetic loci. To prioritise genes underlying the association of these loci with FC, effector gene prediction (EGP) with FLAMES was used[28]. The four loci were mapped to genes *PAX8* (chromosome 2, start and stop base pairs: 113963070:114213070), *EphA3* (3:89451721:90010903), *THBS1* (15:39514832:39764832) and *APOE* (19:45286941:45536941; Fig. 2a, top; "Methods"; Supplementary Data 4, all positions in build GRCh37). The lead variant associated in the *APOE* locus was rs429358, a SNP coding for an exonic missense variant in the *APOE* gene (p.Cys130Arg), which is a well-known common risk variant for AD[29]. The *PAX8* locus was associated at SWS level with 11 edges, *EphA3* with 15, *THBS1* with 4 and *APOE* with 1.

To assess how measurement reliability influences our results, test–retest reliability was calculated in the sample of individuals for whom repeated measurements were available. Edges associated with SWS loci showed higher reliability (median ICC = 0.41, range = [0.20; 0.60]) than the set of all edges (median ICC = 0.19; range = [0.0, 0.60]). Broadly, $h^2_{SNP}$ was correlated with edgewise test–retest reliability (Spearman's $\rho = 0.68$), and improved by applying global signal regression, implying that measurement noise affects the power to detect associated genetic variants differentially (Supplementary Note 3). Consequently, locus yield is likely to increase with increasing sample size.

To evaluate the impact of different phenotypic definitions, we compared our results with GWAS on functional connectivity in prior literature using the UK Biobank. Seventy-seven percent of genome-wide significant (GWS; i.e., $p < 5 \times 10^{-8}$) findings in a study using Independent Component Analysis (ICA)-based edge definition also had GWS associations with phenotypes in this study[10]. Genetic correlation analysis using RSN-level GWAS[8] suggests that edges within a given network are generally more genetically similar to the overall signal of that network, indicating convergence of genetic architecture at different scales of FC analysis (Supplementary Note 5; Supplementary Data 5).

### Genetic loci relevant for functional connectivity have a widespread effect on the connectome

Next, we investigated whether GWS loci associated with FC are expected to have a connectome-wide or edge-specific effect. For this, significance in all GWS loci was examined across all 3321 phenotypes. The significance values across all edges for one locus were used to define the effect extent of a locus ("Methods"). The median effect extent for the four replicated loci was 10% of all edges in the brain (337 edges out of 3321; range: [92; 585]; Supplementary Fig. 2). The *PAX8*, *APOE*, *EphA3*, and *THBS1* loci were estimated to affect 17%, 11%, 9%, and 3% of all functional edges in the brain (585, 368, 306 and 92 out of 3321 edges), respectively. These results indicate these loci have a widespread effect throughout the brain (see Fig. 2b–e for spatial organisation of associations, Supplementary Figs. 3–6), which suggests that the effect of these loci is unlikely to be specific to the edges in which they were detected.

Given FC-associated genes showed a widespread connectome impact, we investigated whether genes implicated in neuropsychiatric disorders show greater effect extent in FC than expected beyond

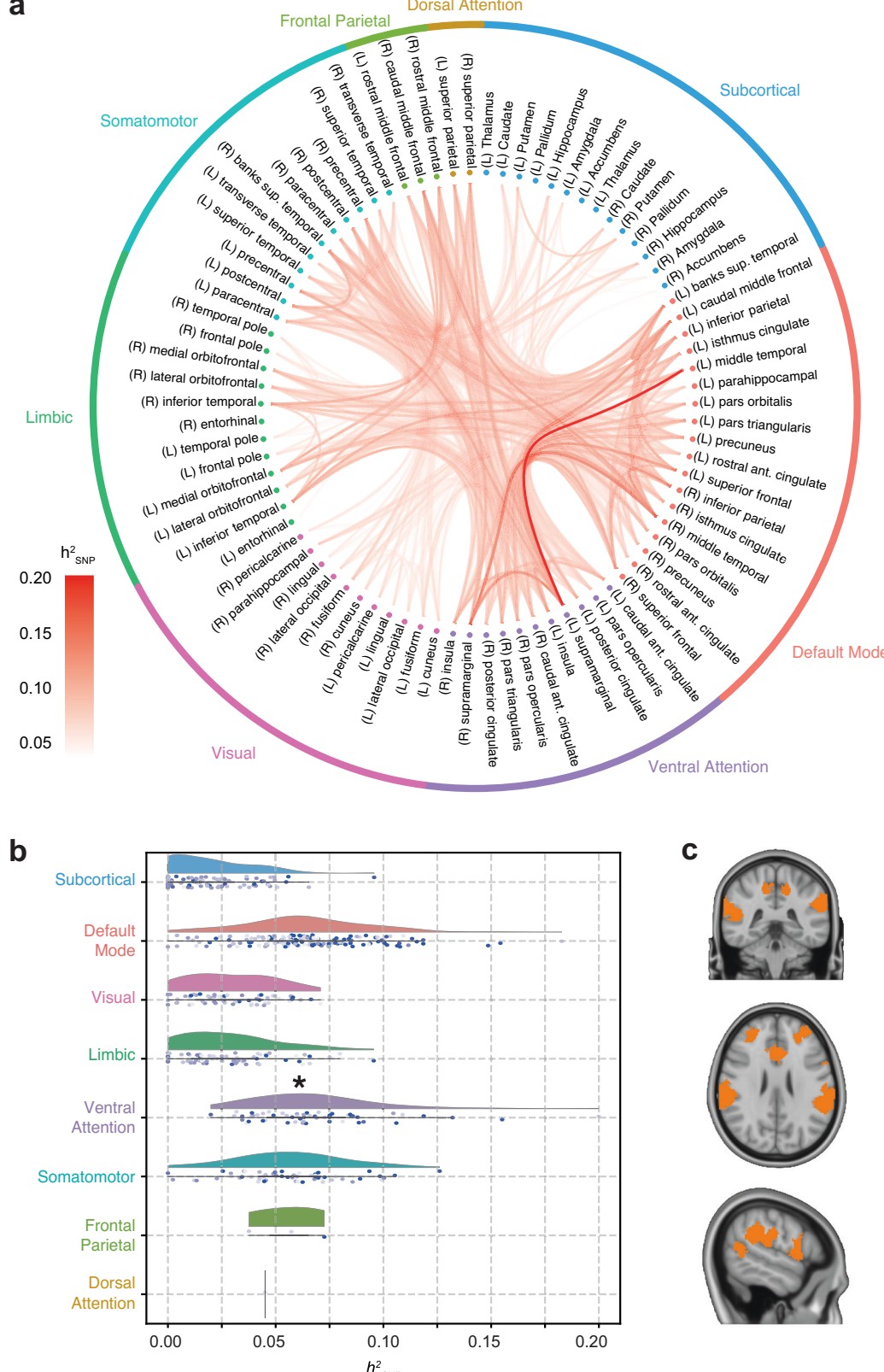

**Fig. 1 | SNP-heritability of the human functional connectome. a** Circos plot of SNP-heritability ($h^2_{SNP}$) estimates across the brain. **b** $h^2_{SNP}$ estimates grouped per network. Half-violin plots represent the distribution of edge $h^2_{SNP}$. Each point below the violin plot is the heritability estimate for each edge belonging to the network. One asterisk (*) represents a *p*-value < 0.05 in a one-sided test for subnetwork enrichment testing (full method description in Supplementary Methods; $p_{\text{Ventral Attention}} = 0.01$; Table 1). The shade of points in the jitter plot depicts the standard deviation of the estimation with lighter points having more uncertainty. **c** Visualisation of the Ventral Attention Network on the brain. RSN parcellation and brain template are available in Data Availability.

highly brain-expressed genes (Supplementary Methods; Supplementary Note 6). We calculated the expected extent of the effect of genes previously associated with seven psychiatric conditions. A significantly larger extent was found for genes linked to anorexia nervosa

($p = 0.03$)[30], Alzheimer's disease (AD; $p < 1 \times 10^{-4}$)[31], and schizophrenia ($p < 1 \times 10^{-4}$)[32]. The gene-sets for AD and schizophrenia remained significant after correcting for multiple testing (one-sided $\alpha = 0.05/7$ disorders tested; Supplementary Fig. 7; Supplementary Data 6).

### Functional connectivity shares a genetic basis with cardiovascular, cognitive and metabolic domains

MAGMA was used to perform genome-wide gene-based association (GWGAS; "Methods") with the goal of investigating whether aggregating small SNP effects within a gene reveals new associations[33]. A GWGAS was run per edge to combine individual SNP $p$-values into gene–edge association scores for a set of 18,852 protein-coding genes. GWGAS revealed associations with 6 unique genes, 4 of which had not been identified in the SNP-based analyses: *EphA3*, *APOE*, *APOC1*, *ZIC4*, *ZIC1*, and *SLC39A12* (two-sided $\alpha_{disc} = 0.05/(18,852 \times 3321)$; Supplementary Data 7). Gene–edge associations with *EphA3* and *SLC39A12* were replicated in the holdout sample (Methods; Supplementary Data 8). *SLC39A12* is a previously unreported gene for FC found to be associated with the connection between the left and right putamen. This gene has not previously been reported in FC GWAS and it is unique to GWGAS, being outside of the four loci discovered in SNP-based testing[10,19,20]. Together with the gene prioritisation results, five

**Table 1 | Heritability enrichment for resting-state networks**

| Subnetwork | $h^2_{SNP}$ | $N_{edges}$ | Enrichment | *P*-value |
|---|---|---|---|---|
| Default Mode | 0.064 ± 0.029 | 153 | 0.62 | 0.29 |
| Dorsal Attention | 0.045 | 1 | 0.21 | 0.42 |
| Frontoparietal | 0.056 ± 0.018 | 3 | 0.20 | 0.42 |
| Limbic | 0.028 ± 0.022 | 66 | -0.85 | 0.80 |
| Somatomotor | 0.057 ± 0.026 | 55 | 0.08 | 0.48 |
| Subcortical | 0.019 ± 0.019 | 91 | -0.82 | 0.79 |
| Ventral Attention | 0.069 ± 0.032 | 55 | 2.28 | 0.01* |
| Visual | 0.029 ± 0.019 | 55 | -1.80 | 0.96 |

$h^2_{SNP}$ represents the average LDSC SNP-heritability estimate of the edges in the subnetwork (mean ± standard deviation). $N_{edges}$ is the number of edges in each subnetwork. Enrichment is the $t$-statistic of the subnetwork enrichment test (Supplementary Methods). One asterisk (*) represents an unadjusted nominally significant $p$-value (i.e., one-sided $\alpha = 0.05$) in subnetwork enrichment testing.

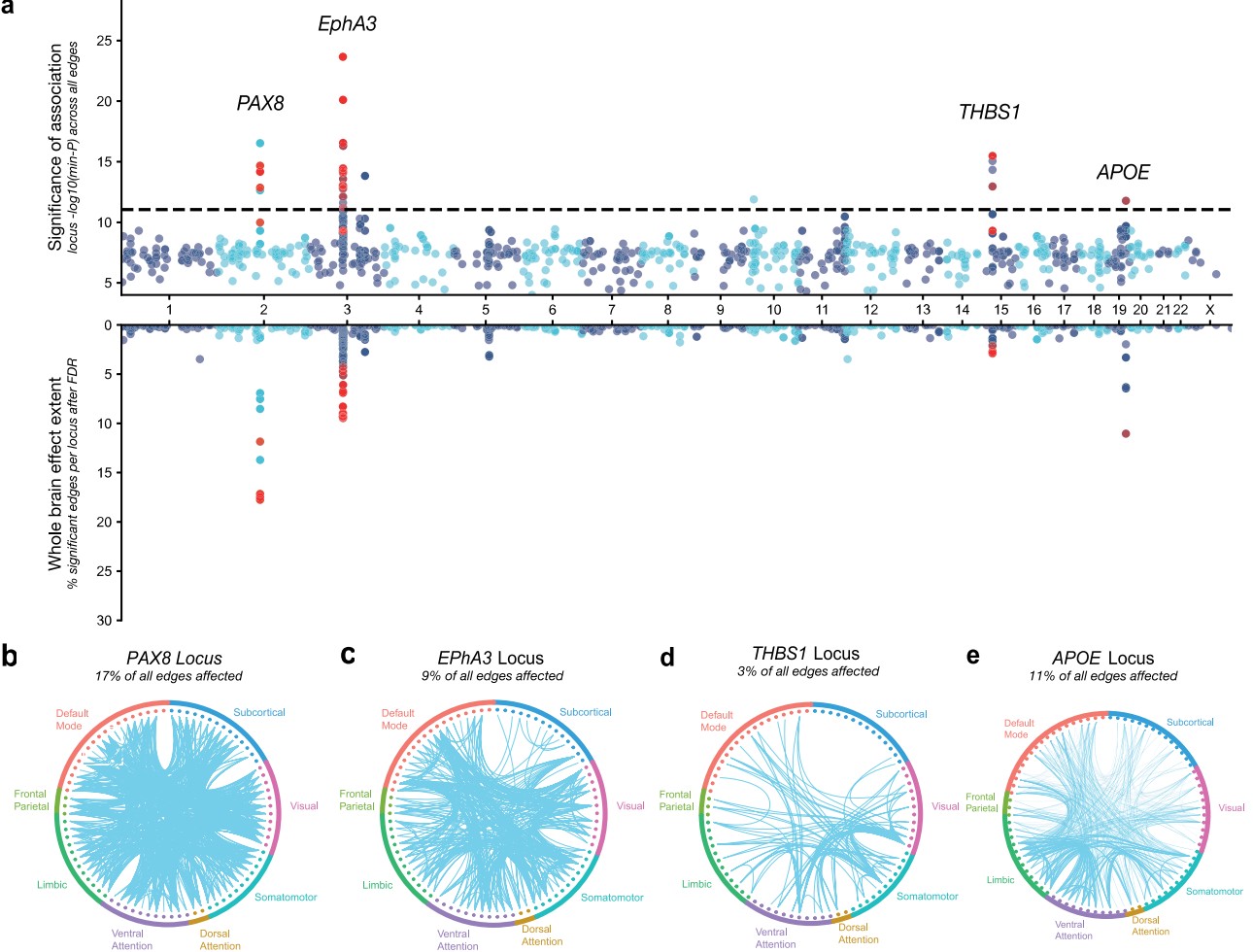

**Fig. 2 | Pleiotropy of genome-wide significant locus–edge associations with functional connectivity. a** Miami plot for locus association and effect extent across 3321 edge-GWAS for edges. A locus for which the lead SNP was found to be GWS significant in at least one edge-GWAS is represented by dots (two-sided $\alpha_{GWS} = 5 \times 10^{-8}$, $N_{discovery} = 24{,}442$ subjects). Replicated locus–edge associations are represented in red (two-sided $\alpha_{replication} = 0.05/208$, $N_{replication} = 3708$ subjects).

Top: -$\log_{10}(p)$ of minimum association $p$-value with the locus across all edges in the discovery sample. Dashed line represents study-wide significance ($\alpha_{SWS} = 5 \times 10^{-8}$/ 3321). Bottom: number of edges that pass FDR correction for locus significance (effect extent estimation). **b–e** Circos plots of the connections predicted to be associated with each replicated locus. Only loci significant for an alpha level FDR-corrected for the number of phenotypes tested are drawn.

genes were found to be associated with FC at SWS and replicated: *EphA3*, with both GWGAS and EGP; *SLC39A12*, with GWGAS alone; *PAX8*, *THBS1* and *APOE*, with EGP alone.

Gene-set enrichment analysis (GSEA) was used to investigate whether genes associated with FC converge on specific biological processes and molecular functions, using hypergeometric testing with GENE2FUNC in FUMA[34]. Two FC candidate gene-sets were analysed: a strict gene-set comprising the five genes with SWS replicated associations and a broad gene-set containing 322 different protein-coding genes associated at GWS level with at least one edge in GWGAS ($\alpha_{disc}$ = 0.05/18,852; "Methods"; Fig. 3a). The strict candidate gene-set for FC was enriched for two gene ontology (GO) molecular functions (protein-lipid complex binding and proteoglycan binding) and nine GO biological processes (top three: tube morphogenesis, nitric oxide mediated signal transduction and regulation of cellular component biogenesis; Supplementary Fig. 8; Supplementary Data 9). Next, the broader candidate gene-set was used to explore more general phenotypic associations using genes for which there is a considerable amount of subthreshold signal. While GWS genes form a bigger gene-set, it is likely to be noisier. However, the broader set of genes was enriched for protein-protein interactions compared to what would be expected for a set of proteins of the same size and degree distribution drawn at random from the genome ($p = 1.0 \times 10^{-6}$; Supplementary Fig. 9), which indicates that the proteins coded by these genes are biologically connected as a group[35]. GSEA revealed 53 processes were significantly associated with FC. These were separated into 10 different domains, namely associations with cardiovascular (9 processes), cognitive (9), metabolic (7), longevity (5), imaging-derived (5), alcohol consumption (4), dietary (3), neurological (3), gastrointestinal (2) and respiratory (2) domains (Fig. 4; Supplementary Fig. 10; Supplementary Data 10). No GO terms, tissue or cell types, or curated gene-sets were significantly enriched for the broad gene-set.

Next, we investigated whether FC was genetically correlated with neuropsychiatric diagnoses using LDSC. Out of all edges, 21 showed a significant correlation with schizophrenia (two-sided FDR-corrected $\alpha$ = 0.05, no association reached Bonferroni significance). No other association survived multiple-testing correction (Supplementary Note 6; Supplementary Fig. 11; Supplementary Data 10). With increasing FC GWAS sample sizes and consequent higher power and better $h^2_{SNP}$ estimations, we expect to find a higher number of significant genetic associations of FC with neuropsychiatric diagnoses. We used mendelian randomisation[36] (MR) to investigate whether there would be evidence of causality between functional connectivity and neuropsychiatric disorders. MR results were inconclusive; this analysis did not have enough statistical power to draw conclusions about bidirectional causality (Supplementary Note 6), underscoring the need for well-powered samples and experimental designs in the assessment of causal links[37].

### Functional interpretation of gene–edge associations points towards neurodevelopmental and angiogenic processes in the genetic control of the functional connectome

The most significant locus–edge association was linked to the *EphA3* gene (minimum $p_{locus} = 2.2 \times 10^{-24}$). This gene is among the top 10% of human genes most intolerant to variation, meaning that mutations are likely to lead to downstream biological consequences[38]. The receptors coded by *EphA3* are involved in neurodevelopmental events such as lobe-specific axonal guidance and cortical expansion of the primate brain[39–41]. The *EphA3* gene has been previously associated in prior GWAS of language-related FC phenotypes and structural connectivity[20,42]. This work extends the prior association of this gene with structural connectivity beyond functional language circuits, showing a widespread association with FC.

The most frequent genetic association was linked to the *PAX8* gene on chromosome 2 (minimum $p_{locus} = 2.9 \times 10^{-17}$, effect extent =

17% of all edges). This master regulator transcription factor gene has been associated with several FC phenotypes, from graph measures[21] to RSNs[8] and individual edges in the UK Biobank cohort[20]. It has been related to hyperthyroidism and plays a key role in epithelial cell survival and proliferation[43–45]. The genetic signal colocalised in a single variant associated with this gene for all edges with which it was associated, showing a highly localised genetic influence.

The *APOE* and *THBS1* genes were found to drive the enrichment of FC for biological processes of proteoglycan binding (enrichment $p = 3.57 \times 10^{-5}$) and protein-lipid complex binding (enrichment $p = 2.0 \times 10^{-5}$; Supplementary Fig. 7; Supplementary Data 9), which are both important in the genetic aetiology of AD[31,45–47]. The *APOE* gene codes for apolipoprotein E, which is mostly synthesised by microglia and astrocytes in the brain and plays a fundamental role in cholesterol metabolism[46]. It has been consistently related to longevity[46] and heart health[48]—phenotypes genetically associated with FC in our study (Fig. 3b). *THBS1* is a gene coding for thrombospondin-1, a glycoprotein found in the extracellular matrix involved in postnatal neuronal migration[49]. We found enrichment for several biological processes related to angiogenesis, which were driven by association with these genes (e.g. tube morphogenesis and development, negative regulation of blood vessel endothelial cell migration and regulation of sprouting angiogenesis; see Supplementary Fig. 5 and Supplementary Data 9). The lifetime expression of these genes is different: *APOE* is highly expressed in the brain postnatally until late adulthood, whereas the peak expression of *THBS1* is perinatal (Supplementary Figs. 12 and 13). The *APOE* gene has also been specifically associated with changes in brain morphology across the lifespan[13]. Our findings indicate a diverse genetic modulation of FC throughout the lifetime, with a neurodevelopmental and an older age component.

A whole gene–edge association with FC was found between *SLC39A12* and connectivity in the bilateral putamen (minimum $p_{gene} = 2.7 \times 10^{-11}$; Supplementary Data 7). This gene is differentially expressed in astrocytes and the choroid plexus[50]. It is involved in the transport of zinc from the extracellular space to the cytoplasm across the cell membrane, implicated in the processes of neurulation and neurite extension, and shown to have an altered expression profile in schizophrenia[51,52]. *SLC39A12* has previously been associated with anatomic alterations in the putamen, among other cortical and subcortical brain regions (T1 and T2*-imaging)[9,12,18]. These changes were hypothesised to be associated with iron deposition in ageing and pathology[12]. Rare predicted loss-of-function variants in this gene have no other reported associations with imaging or neuropsychiatric phenotypes[53]. These results suggest that S*LC39A12* might underlie variability in both imaging and psychiatric phenotypes.

## Discussion

Coordinated activity across brain regions is fundamental to cognition and behaviour[2]. Genetic variation explains individual differences in these large-scale neural dynamics and can point towards biological processes underlying this variability. Here, we investigated the biological underpinnings of the human functional connectome by performing GWAS analysis of functional brain connectivity. We directly map genetic effects onto specific areas of the brain and quantify their extent from localised circuit-level associations to widespread pleiotropic influences. Individual edge-wise GWAS were mapped and summarised using gene-based mapping techniques and showed that genetic associations with FC are shared with both physiological and cognitive traits, placing the functional connectome in the intersection between brain and physical health.

We adopted an edge-wise anatomically-based parcellation of the brain to investigate the genetics of FC, allowing the discovery of the strongest genetic effects with high anatomical precision. This level of analysis enabled us to interrogate the role of genetic variation in individual differences in FC at several levels of macroscale connectivity

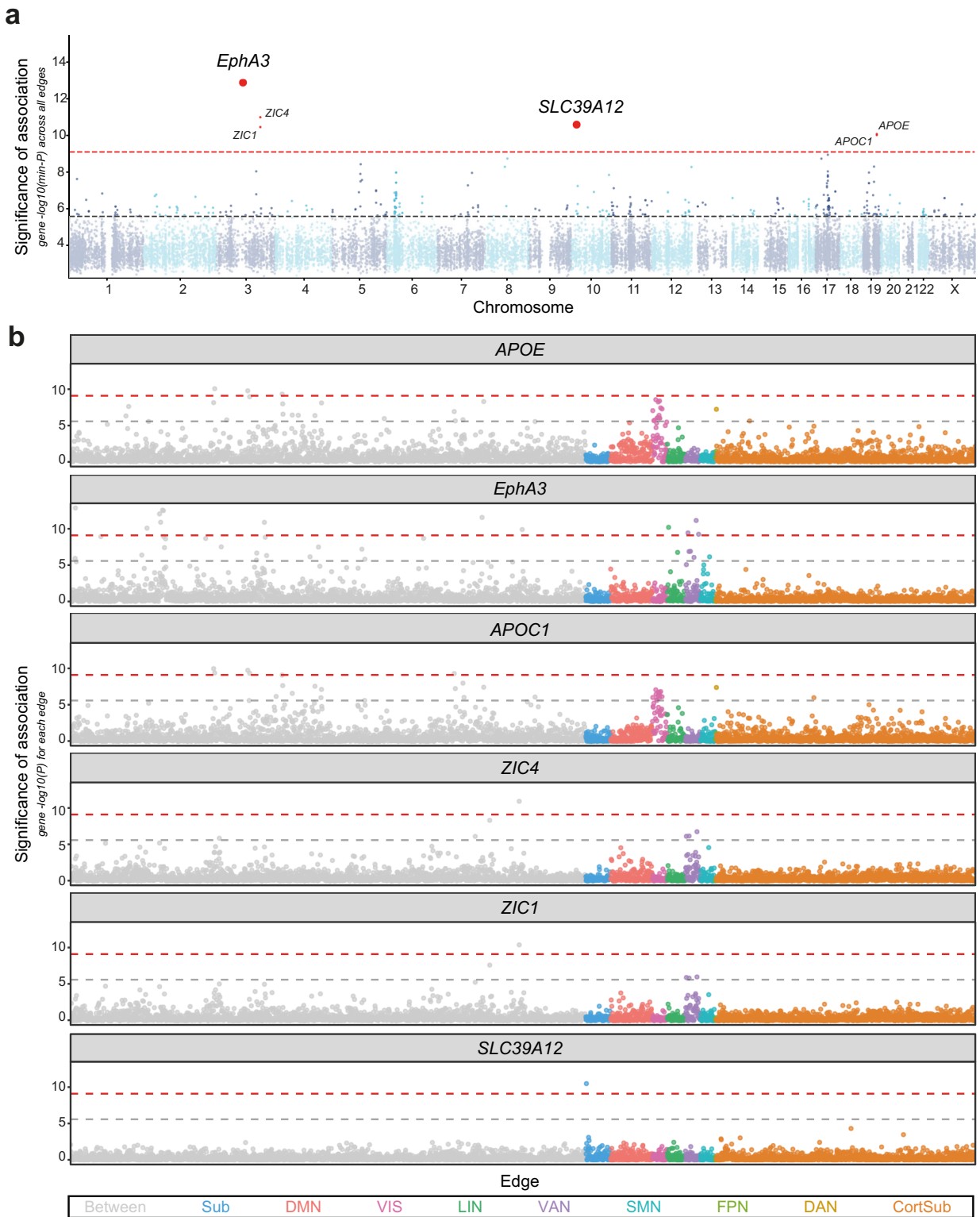

**Fig. 3 | Gene-based testing of functional connectivity. a** Manhattan plot for GWGAS across 3321 functional brain connections. Each dot represents one protein-coding gene. The height of the point depicts the $-\log_{10}(p)$ of minimum gene association $p$-value with all the phenotypes, i.e. the strongest statistical association of each gene across all phenotypes. Dashed lines represent SWS (two-sided $\alpha_{SWS} = 0.05/(18,852 \times 3321)$, $N_{discovery} = 24,442$ subjects, in red) and GWS (two-sided $\alpha_{GWS} = 0.05/18,852$, in grey) for gene-based testing. Genes with an SWS minimum $p$-value are in red. *EphA3* and *SLC39A12* were replicated in a holdout sample (two-

sided $\alpha_{replication} = 0.05/6$, $N_{replication} = 3708$ subjects). **b** Univariate gene–edge association $p$ values for the 6 SWS genes discovered through GWGAS. Each point represents an edge, and the colour of the point indicates the subnetwork to which it belongs. Between Edge between networks, Sub Subcortical, DMN Default Mode Network, VIS Visual Network, LIN Limbic Network, VAN Ventral Attention Network, SMN Somatomotor Network, FPN Frontoparietal Network, DAN Dorsal Attention Network, CortSub Cortico-subcortical edge.

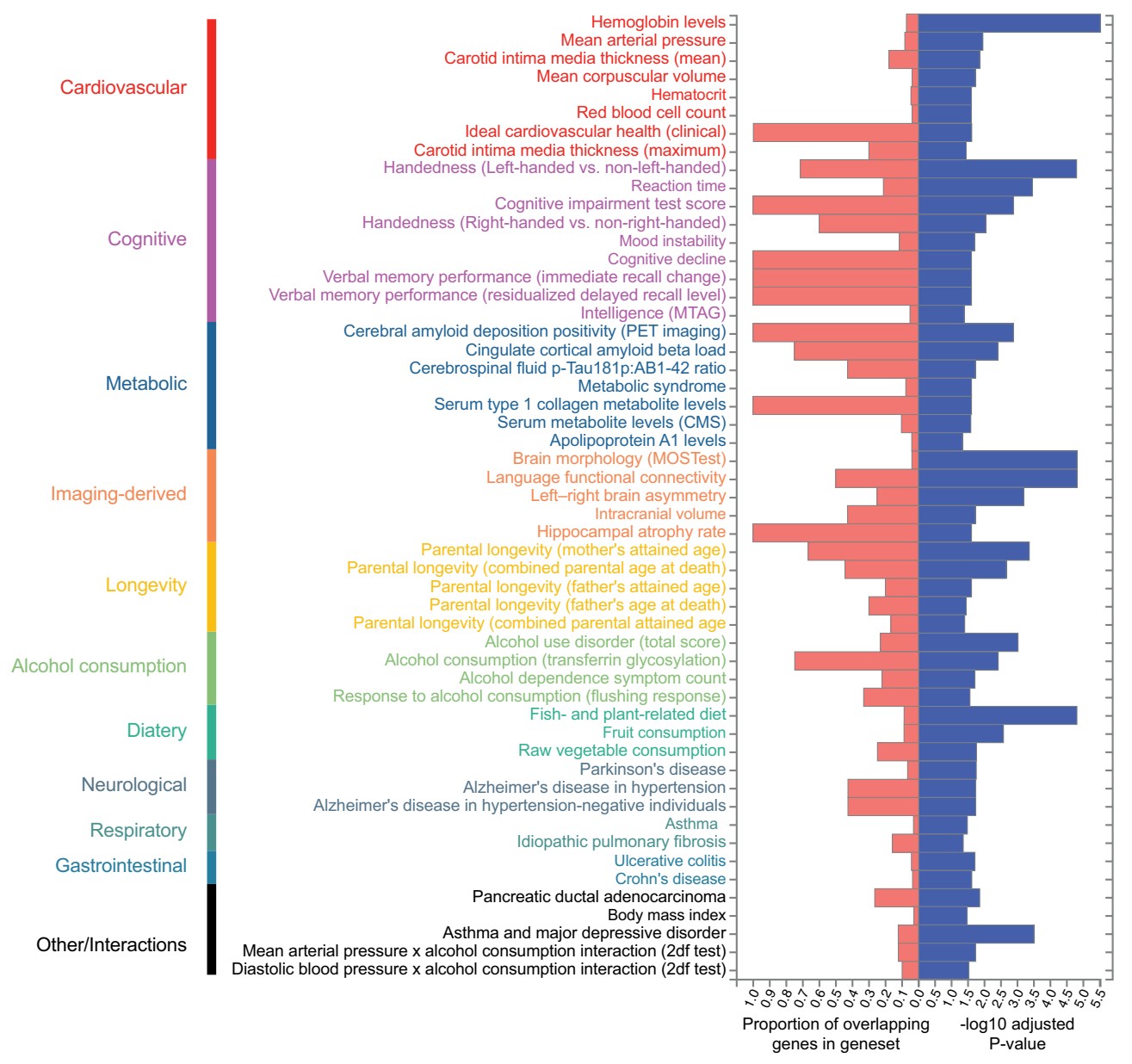

**Fig. 4 | Gene-set enrichment analysis for all annotated genes ($N_{genes}$ = 322).** Significantly enriched gene-sets were manually sorted by domain.

organisation—individual edges, modules, and resting-state networks. Each edge in this manuscript represents the functional coupling between two brain regions, which not only reflects local interactions between two brain regions but also their participation in large-scale network and physiological processes[54,55]. Consistent with this interpretation, we find that the strongest loci were expected to influence many edges rather than a small, highly specific subset, indicating the effects of common genetic variation in FC are widespread and pleiotropic.

Family studies[6,56] and GWAS[8,10] have reported resting-state networks (RSNs) to be heritable, but the heritability of all functional links of the human connectome remained unknown. We find that over 30% of edges in the human functional connectome are significantly heritable, explaining on average 6.5% of individual variation in FC. This number is comparable to the 9.1% found for structural connectivity using comparable methods[42].

Four genes—*PAX8*, *EphA3*, *THBS1*, and *APOE*—particularly stand out. These genes have been previously shown to influence macroscale brain connectivity, with *EphA3* playing a role in both structural and functional connectivity[8,11,20,22,42]. This supports the idea of high pleiotropy between MRI-derived measures and neuropsychiatric traits[17,18,57,58] Despite their known genetic associations with various FC measures[8,12,20], *APOE*, *PAX8*, and *THBS1* were not linked to edge-level structural connectivity in recent large-scale GWAS[42,59], indicating these might specifically influence FC. In addition to these four genes, we found a whole-gene association of FC with *SLC39A12*. To our knowledge, *SLC39A12* previously only had been associated with brain volumetric features and not with brain connectivity[9,12,18]. *SLC39A12* encodes a zinc transporter involved in metal ion homoeostasis, which has been implicated in neurodevelopment and synaptic function[51,52]. This finding seems to be specific to the putamen, which is consistent with previous literature[9,12,18]. This suggests that distinct molecular mechanisms may also have a localised effect.

Functional annotation of the set of putative FC genes revealed associations with cardiovascular, cognitive and metabolic phenotypes. Cardiovascular health has been shown to play an important role in the maintenance of FC, both phenotypically[60–62] and genetically[22]. We find cognitive phenotypes to be genetically associated with FC, connecting

the genetic factors of FC and those of cognition. Prior GWAS of brain networks specific to schizophrenia and bipolar disorder showed that connectomic alterations in these conditions are associated with the Notch signalling pathway, which is involved in neural development and blood vessel formation, among other processes[63]. This suggests a role of cardiovascular health and FC in supporting healthy cognition, with angiogenesis being a putative mechanism for the relationship between these three domains. Moreover, enrichment for phenotypes in the metabolic domain, together with the association with *APOE*, suggests a link between the genetic factors influencing FC and AD. In particular, genetic associations between FC and cerebrospinal fluid p-Tau181-p:AB1-42 ratio, cerebral amyloid deposition positivity, apolipoprotein A1 levels and cingulate cortical amyloid beta load point towards important processes in the pathophysiology of AD[64–66]. These findings highlight extensive pleiotropy between brain connectivity and traits from several phenotypic domains, indicating an interplay between cognition, functional brain connectivity and physical health on a genetic level.

This study has limitations to consider when interpreting its findings. First, a mass-univariate approach entails a substantial multiple-testing burden, which reduces power and may increase false negative rate. As a result, study designs such as the one performed here are more attuned to detecting pleiotropic genetic effects, as genetic variants influencing more connections will have an increased probability of being identified for at least one edge. This is of note in our context where different signal-to-noise ratios imply differential power to detect genetic effects on each edge, as shown by the higher test–retest reliability of edges with SWS associations. Consequently, locus yield and SNP-heritability estimates should be interpreted within the context of the current power level and the sensitivity to detect genetic effects across the brain. In particular, the specificity of each finding should be interpreted cautiously as SWS loci were shown to be expected to have a widespread effect on the brain. Next, the use of a single cohort of individuals of older adults with European ancestry may limit the generalisability of our findings, particularly considering the extensive literature in imaging-genetics using the UKB as a study cohort[67]. Replication in a sample with different demographics (particularly of younger age and diverse ethnic and socioeconomic backgrounds) may clarify the effects of genetic variants across the lifetime and associations with relevant clinical outcomes. Studies in cohorts beyond the UK Biobank and experimental design setups are crucial to further assessing the validity of the link between the genetic bases of FC and neuropsychiatric disorders as well as its utility as a clinical biomarker. Third, despite comparing different methods of extracting functional connectivity, the impact of choice of atlas or of dynamic connectivity measures has not been examined in this study. Resting-state fMRI may be sensitive to motion[68], arousal[69], physiological noise[70], and preprocessing choices[71]. However, these would mainly have transient state effects on FC and reduce our power to detect genetic associations. Here, we find a set of biologically connected genes with associations with phenotypes of physical and cognitive domains, highlighting the trait nature of macroscale brain connectivity. Future work investigating other functional connectivity metrics, such as dynamic functional connectivity, could provide further insights into the state-dependent component of functional connectivity and facilitate the study of how genes interact with environmental factors. Finally, the small size of the strict candidate gene-set makes GSEA sensitive to chance effects. The statistical significance of the highlighted biological processes should be interpreted with caution. However, the broader GSEA indicates convergence, which supports our interpretation of the results.

In this work, we contextualise the genetic basis of the human functional connectome in relation to physical and neurocognitive traits. Our findings show that brain connectivity and phenotypes from physical and cognitive domains share a common genetic basis, extending pleiotropy across functional brain connections to other dimensions of physical and cognitive health.

## Methods
### Study cohort
Genetic and neuroimaging data from the UK Biobank (UKB), a large-scale cohort with neuroimaging, genotyping, clinical, demographic, and behavioural data of volunteer participants from a population-based sample of older adults in the UK were used in the present study[23]. The UKB initiative has been approved by the National Research Ethics Service Committee (reference 11/NW/0382) and data were accessed under application number 16406. Joint neuroimaging and genotypic data (imputed from Single Nucleotide Polymorphism (SNP) arrays) were available for a total of 40,682 subjects. From those, 5000 subjects were randomly assigned to a holdout sample before any analysis for replication purposes. Subjects with sex aneuploidy, discordant reported and chromosomal sex, high degrees of relatedness ($N = 5241$) and non-European ancestry ($N = 2034$) individuals were excluded[72]. After quality control, a total of 24,451 subjects in the discovery sample (13,152 inferred chromosomal females) and 3708 (2046 females) in the holdout sample were included in our analysis. Median age of included participants is 64 years and 9.4% of subjects are left-handed. A detailed overview of exclusions and a full description of the demographic characteristics of the discovery and holdout samples are provided in Supplementary Note 1.

### Neuroimaging and genotypic data preprocessing
UK Biobank imputed genotyping SNP-array data were used for this study (in build GRCh37). The imputation and quality control performed by the UK Biobank team are described extensively elsewhere[23]. After additional in-house genomic quality control (Supplementary Methods), a total of 9,380,668 high-quality SNPs were hard-called under a certainty threshold of 0.9[73]. The acquisition and preprocessing protocols for the neuroimaging data are described in the UKB Brain Imaging documentation[74]. Resting-state functional magnetic resonance imaging (rs-fMRI) was used with T1 surface model files and structural segmentation from FreeSurfer (v6.0). Functional connectivity was computed using CATO (Connectivity Analysis Toolbox; v3.1.6)[75]. Coregistration was performed by aligning the subject average of rs-fMRI across all time points with the T1 image[76]. Next, surface-based cortical parcellation was performed using FreeSurfer parcellating the cortical mantle according to the Desikan-Killiany atlas and subcortical volumes were automatically segmented into the aseg atlas in the T1 image[77,78]. This resulted in a whole brain parcellation of a total of 82 brain areas (68 cortical + 14 subcortical). Motion metrics, alongside their first-order drifts and their linear trends, and the average signal of voxels in both cerebrospinal fluid and white matter were regressed out of the rs-fMRI signal. Global signal regression (removal of the mean whole-brain BOLD signal to reduce global noise) was applied. A zero-lag bandpass filter ([0.01−0.1] Hz band) and motion-scrubbing were applied to the time series: frames with more than two violations were discarded along with their closest backward neighbour (maximum framewise displacement = 0.25, maximum DVARS = 1.5)[79].

### Phenotype reconstruction
FC matrices were constructed by calculating pairwise Pearson's correlations between the average preprocessed time series of each of the 82 parcellated brain areas. Subjects for which average head motion parameters (across space and time), signal-to-noise ratio or discrepancy between T1-weighted and fMRI scan deviated from the median more than five times their median absolute variation were excluded from the analysis[80].

Each individual element of the FC matrix across all subjects was used as a phenotype for a SNP-based GWAS. Phenotypes were standardised across subjects and residualised for total intracranial volume.

## SNP-based GWAS

Identification of common genetic variants involved in FC was carried out using PLINK2.0[81]. A total of 3,321 SNP-based GWAS on single-edge connectomic data were performed for 8,790,386 imputed and genotyped SNPs. This analysis was performed on independent (Linkage Disequilibrium threshold, $r^2 > 0.1$), common (UKB MAF > 0.01), and genotyped SNPs or SNPs with very high imputation quality (INFO > 0.9). The first 20 principal components (extracted using FlashPCA2[82]; Supplementary Methods) were used as covariates in all GWAS together with sex, age, handedness, genotype array, and covariates specific to fMRI: scanning site, time to echo, table coordinates, coil position, signal-to-noise ratio, mean head motion, intensity scaling parameters and framewise displacement[80]. The full set of covariates was standardised.

## SNP-heritability estimation

Using linkage disequilibrium score regression (LDSC; v.1.0.1)[24], SNP-based heritability for each edge-GWAS was estimated. SNPs in the HAPMAP3 reference panel were included (Data Availability), leaving 1,137,297 SNPs for analysis. The significance level for heritability was defined as $\alpha_{heritability} = 0.05$. Additionally, a cut-off $\lambda > 1.02$ was set to determine which phenotypes had enough polygenic signal to run LDSC and for the LDSC intercept (<1.1) to control confounding from population stratification[24].

A linear regression model with a fixed variance-covariance structure of the residuals was used to determine whether any of the seven Yeo-Krienen RSNs[1] (and subcortical network) was enriched for the $h^2_{SNP}$. The genetic covariance matrix was estimated through the calculation of edge-wise correlations between each pair of edges for all the individuals allocated to the discovery sample. The significance level was set at $\alpha_{enrichment} = 0.05/8$ to correct for multiple testing for the seven RSNs and subcortical network. See Supplementary Methods for more details on the method.

## Definition and effect extent of genomic risk loci

The identification of independent genomic loci associated with each FC trait was done using the linkage disequilibrium-based result clumping procedure implemented in PLINK1.9[81]. Independent significant genome-wide (GWS) SNPs were defined as those with $p < 5 \times 10^{-8}$ for each trait in our discovery sample. An additional multiple testing correction was applied to account for the number of traits: if SNPs were associated at $p < 5 \times 10^{-8}/3321$, these were considered study-wide significant (SWS). An LD panel derived from 10,000 unrelated UKB individuals was used for this analysis. Then, we clumped together all variants with $p < 0.01$ that are in LD with (LD threshold, $r^2 > 0.9$) and in a radius of 250 kb from each independent significant SNP. The most significant SNP (i.e., lowest $p$-value) is referred to as the "lead SNP" of the locus. A locus–edge association refers to each [edge; lead SNP] pair for which the lead SNP was found to be genome-wide significantly associated with the functional connection. Only SWS loci were mapped onto genes.

A total of 1490 locus–edge associations (864 unique lead SNPs) were discovered at GWS level and further assessed across all brain connections to estimate the effect extent of these loci throughout the brain. For each of these 864 loci, the $p$-values for all SNPs were combined using a SNP-wise mean model in MAGMA v1.10[33], resulting in one $p$-value per locus for each edge. Benjamini-Hochberg FDR correction for the number of traits (3321) was applied across edges to these $p$-values to quantify with how many traits each locus was expected to be associated. The ratio between the FDR-significant and total number of tests is referred to as the effect extent of a locus. The effect of trait dependencies and $p$-value thresholding among significant edges in the metric was investigated and the metric was found to be robust to these (Supplementary Methods).

## Genome-wide gene association studies

Genome-Wide Gene Association Studies (GWGAS) was used to detect (whole) gene–edge associations in the functional connectome. A gene–edge association refers to each [edge; gene] pair for which the gene was found to be associated at SWS with the respective functional edge. Each SNP-based GWAS summary statistics was used to perform a GWGAS in MAGMA[33]. A SNP-wise mean model was applied to the summary statistics using a reference of 10,000 individuals from our sample to test the joint association of all SNPs within 18,852 protein-coding genes with each trait. A multiple-testing corrected significance threshold $\alpha_{gene,disc} = 0.05/(18,852 \times 3321)$ was used.

## Replication

Replication of the discovered locus–edge and gene–edge associations was performed in the holdout sample of 3708 individuals. A locus–edge association was considered replicated if the lead SNP in that locus for that edge (i) had a concordant effect direction in the discovery and replication samples, (ii) had or was in LD ($r^2 > 0.9$) with a SNP of MAF > 0.01 in the replication sample, and (iii) was Bonferroni-significant at $\alpha_{SNP,rep} = 0.05/208$ (208 SWS locus–edge associations discovered). A gene–edge association was considered replicated if the gene was Bonferroni-significant for that edge in the replication sample at a $\alpha_{gene,rep} = 0.05/6$ (6 SWS genes tested).

## Prioritisation of genes and functional annotation

Two loci from two different GWAS were considered to overlap if their respective lead SNPs were separated by less than 250K base pairs. These loci were validated by co-localising the different GWAS with coloc (v5.2.2; Supplementary Note 4, Supplementary Data 3)[83]. Estimation of the LD reference panel consisting of all the individuals in each discovery GWAS was carried out on LDStore2 (v2.0)[84]. The resulting LD reference panel was used to fine-map each of the overlapping loci with FINEMAP (v1.4.1)[85], using a threshold of a maximum of 10 causal variants per locus. The effector gene prediction (EGP) for each credible set was carried out using FLAMES (Fine-mapped Locus Assessment Model of Effector geneS; v1.0.0) with a 750 kb window on each side of the centroid of the credible set and prediction of coding genes[28]. The Polygenic Priority Scores (PoPS; v0.2) for MAGMA were calculated using the pathway-naïve feature set provided by the authors (Data Availability)[86]. Predictions with an FLAMES score above 0.05 were kept for further analyses.

Pathway convergence of the gene-sets resulting from univariate gene-based testing was investigated using GENE2FUNC in FUMA (Functional Mapping and Annotation of Genome-Wide Association Studies)[34]. GENE2FUNC performs hypergeometric tests to test if genes of interest are overrepresented in any of the pre-defined gene sets. A broad gene-set containing all associations across all GWGAS + EGP (322 genes) and the gene-set of replicated genes (1 + 4 genes) were tested separately. Enrichment testing was performed using a background of all protein-coding genes. Pathway-specific enrichment was calculated for gene-sets from Gene Ontology molecular functions, cellular components and biological processes, and curated gene-sets from MSigDB (gene-set categories C2 and C5; v7.0) and all gene-sets in the GWAS catalogue[87,88]. Gene-sets where at least two of our input genes featured were considered. Benjamini-Hochberg multiple-testing correction per gene-set category for the hypergeometric gene-set enrichment testing and tissue differential expression analysis was adopted.

## Reporting summary

Further information on research design is available in the Nature Portfolio Reporting Summary linked to this article.

# Data availability

The genome-wide summary statistics data generated in this study can be accessed through https://doi.org/10.5281/zenodo.18429460[89]. The

imaging and genotyping data are protected and are not available due to data privacy. The Source data used to plot each figure are provided with this paper. The following data have been used to perform the analyses on this manuscript: LD reference for LDSC, https://www.internationalgenome.org/category/reference/; RSN Annotation Files, https://surfer.nmr.mgh.harvard.edu/fswiki/CorticalParcellation_Yeo2011; MNI template: https://nist.mni.mcgill.ca/mni-average-brain-305-mri/; FLAMES pathway-naïve feature set: https://zenodo.org/records/10409723; Case-control disease sumstats: https://pgc.unc.edu/for-researchers/download-results/. Source data are provided with this paper.

## Code availability

All used software tools and code are publicly available: CATO, http://www.dutchconnectomelab.nl/CATO/; FUMA, http://fuma.ctglab.nl/; MAGMA, https://ctg.cncr.nl/software/magma; LDSC, https://github.com/bulik/ldsc; PLINK, https://www.cog-genomics.org/plink/; FlashPCA2, https://github.com/gabraham/flashpca; FLAMES, https://github.com/Marijn-Schipper/FLAMES; FINEMAP & LDStore2, http://www.christianbenner.com/; coloc R package, https://chr1swallace.github.io/coloc/.

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

## Acknowledgements

B.A.P.C.M., D.P., C.R., M.P.vd.H., and M.S. were funded by a Dutch Research Council (NWO) Gravitation grant: BRAINSCAPES: A Roadmap from Neurogenetics to Neurobiology (grant no. 024.004.012 [to D.P.]). D.P. and C.d.L. were funded by a European Research Council advanced grant (grant no. ERC-2018-AdG GWAS2FUNC 834057 [to D.P.]). M.P.vd.H. and K.H. were funded by an European Research Council Consolidator grant: CONNECT (grant no. 101001062 [to M.P.vd.H]). M.P.vd.H. was funded by a Dutch Research Council (NWO) VICI grant: BrainDiversity (grant no. VI.C.241.074). J.E.S. is supported by the Dutch Research Council (NWO) VENI Grant 201G-064. This manuscript was submitted as a preprint to medRxiv. This research has been conducted using the UK Biobank resource under application 16406. We thank the numerous participants, researchers, and staff who collected and contributed to the data. We thank SURF for the support in using the National Supercomputers Snellius and LISA. Part of the work in this manuscript was carried out with the support of the SURF Cooperative using grant EINF-14888. We thank Dr Rachel M. Brouwer, Dr Sophie van der Sluis and Ilan Libedinsky for all the feedback. Finally, we would also like to acknowledge Dr Elleke P. Tissink and Dr Siemon C. de Lange, who preprocessed the MRI data.

## Author contributions

B.d.A.P.C.M.: Conceptualisation, Methodology, Formal Analysis, Writing, Visualisation. M.S.: Formal Analysis, Feedback. C.R.: Visualisation, Feedback. C.d.L.: Methodology, Feedback. K.H.: Feedback. D.P.: Funding acquisition, Feedback. J.E.S.: Methodology, Supervision, Feedback. M.P.v.d.H: Conceptualisation, Methodology, Resources, Supervision, Project Administration.

## Competing interests

M.P.v.d.H. works as a consultant for Hoffman-La Roche and is part of the editorial team of Wiley Human Brain Mapping; they had no role in this study. The authors declare no competing interests.
