## [Transparent Peer Review file · Nature Communications]

The genetic landscape of human functional brain connectivity

Corresponding Author: Professor Martijn van den Heuvel

Version 0:

Reviewer comments:

Reviewer #1

(Remarks to the Author)

This study presents a large-scale genome-wide association study (GWAS) of pairwise functional brain connectivity (FC) across 3,321 interregional connections in ~28,000 participants from the UK Biobank. The authors identify several loci and genes (e.g., EphA3, PAX8, APOE, SLC39A12) with pleiotropic effects on FC, and demonstrate genetic overlap with neuropsychiatric, cardiovascular, and cognitive traits.

While the study is ambitious and methodologically rigorous, I have concerns regarding its interpretability, and application. Major:

1. Previous studies have already investigated the genetic architecture of functional brain connectivity (e.g., Zhao et al., 2022). In contrast to these approaches, which integrate multiscale network-level traits, this study performs mass-univariate GWAS across over 3,000 pairwise edges. While this design is fine-grained, it also introduces an enormous multiple testing burden, likely diluting statistical power and inflating false negatives. The authors should consider comparing their findings directly with those from Zhao et al., and/or apply multivariate GWAS techniques (e.g., MOSTest, Sha et al. Science Advance, 2023) to validate their key results. Additionally, the generally low heritability values across traits deserve further explanation.
2. The proposed “effect extent” score is an interesting concept for quantifying the breadth of gene influence across traits, but it is currently presented without sufficient statistical justification. The metric does not appear to correct for phenotype correlations, or the inherent arbitrariness of the p-value thresholding used to define “significant” edges. Without replication or statistical modeling of trait dependency, this measure should be interpreted with greater caution.
3. The brain is inherently organized at multiple scales (e.g., modules, networks, gradients). This study focuses exclusively on edge-level analysis, without integrating these higher-order structures. As a result, it is difficult to interpret the biological relevance of the findings within the broader context of brain functional architecture.
4. The discussion of pleiotropic genes is largely descriptive and lacks quantitative metrics to assess the degree of pleiotropy across edges. The authors could improve this by applying phenotypic clustering, genetic correlation matrices, or network-based enrichment analyses to identify modules or subnetworks consistently influenced by shared genetic variation.
5. Although some genes implicated in FC are also associated with major brain disorders, the manuscript stops short of establishing clear mechanistic pathways. Simply observing overlapping loci or tissue enrichment does not sufficiently establish FC as an intermediate phenotype. A more systematic investigation—such as polygenic risk scoring (PRS), LDSC-based genetic correlation, or trait-pathway mediation analysis—would considerably strengthen this claim.
6. While the authors identify genetic variants associated with FC, they do not test causality or directionality. Incorporating Mendelian Randomization (MR), causal mediation models, or network-based gene prioritization could clarify whether SNPs influence clinical traits via connectivity mechanisms. Without such analyses, the study remains largely descriptive and falls short of positioning FC as a valid biomarker for brain disorders.

Minor

1. In Figure 1c: the authors claim enrichment of heritability in the Ventral Attention Network; however, this does not survive multiple testing correction. The rationale for labeling asterisks differently than in Table 1 should be clarified. In Figure 2a: Consider adding a horizontal line to indicate the significance threshold. In Figures 2b–e: Please indicate the number of significant edges for each panel to enhance comparability.
2. The manuscript applies a single atlas to define FC edges but does not discuss the potential impact of atlas choice on results. A brief comment on robustness across parcellations would be helpful.
3. Ensure gene names are consistently italicized throughout the manuscript (e.g., EphA3, PAX8, SLC39A12), as required by

scientific conventions.

Reviewer #2

(Remarks to the Author)

This manuscript investigates the genetic architecture of functional brain connectivity using data from the UK Biobank. The authors identified 72 loci associated with pleiotropic effects observed across the brain and involved in functional pathways related to neurodevelopment, cardiovascular, and cognitive phenotypes. Overall, this study has strong potential to greatly improve our understanding of the genetic basis of individual differences in functional brain connectivity. Below are several suggestions for improvement.

Introduction

The authors state that previous studies have demonstrated high heritability of functional networks. However, this appears inconsistent with the findings of Elliott et al. (2018, Nature), which did not report strong heritability for functional connectivity measures in the UK Biobank. This discrepancy could be better discussed.

The manuscript emphasizes "pleiotropic genetic signals across brain connections." Did the authors also investigate non-pleiotropic / connection-specific genetic effects?

I am not sure why identifying pleiotropic loci associated with FC will help in understanding brain-related disorders. Could you clarify?

Methods

Global signal regression was not applied. Given the ongoing debate about its effects on functional connectivity measures, the authors should consider including sensitivity analyses using GSR. More broadly, running multiverse analyses (e.g., testing alternative preprocessing strategies) would help assess the robustness of the reported genetic associations (see the conclusion of this paper <https://www.nature.com/articles/s41586-020-2314-9>).

The manuscript reports the use of 82 regions from the Desikan-Killiany and ASEG atlases, yet the results refer to the Yeo 7-network parcellation. I am confused here. Could you justify your choice of atlas, particularly in light of more commonly used functionally derived alternatives such as the Schaefer-400 or Yeo-177 parcellations?

Replicating or extending the findings in an independent dataset with distinct demographic characteristics (e.g., ABCD) would greatly strengthen the generalizability and robustness of the results. It would be particularly interesting to assess whether key findings, such as the APOE association, replicate across cohorts.

The authors focus exclusively on static functional connectivity. Could they elaborate on the rationale behind this choice and discuss the potential for future work using complementary metrics, such as dynamic connectivity, network topology, or graph-theoretical features?

It would be helpful to clarify how the edges were defined (the manuscript does not appear to include any graph-theoretical analyses).

Version 1:

Reviewer comments:

Reviewer #1

(Remarks to the Author)

Thank you very much for the detailed and thoughtful response, as well as for performing the additional comparison. I appreciate the effort that went into these revisions.

However, I still have a few remaining concerns that I would like the authors to address more explicitly in the manuscript.

1. Novelty relative to Zhao et al.

From your new analysis, a large proportion of the genome-wide significant (GWS) loci reported by Zhao et al. are replicated in your dataset (you report ~70–77% replication, depending on the definition), and one third of your own GWS loci overlap with Zhao et al. within 250 kb. At the same time, many of your strongest loci (including those reaching study-wide significance) map to genes such as PAX8, EphA3, THBS1 and APOE that have already been implicated in functional connectivity or related imaging phenotypes in previous UK Biobank studies.

As a reader, this makes it somewhat difficult to understand what is truly new about the present work beyond (i) adopting a finer-grained edge-wise parcellation and (ii) re-detecting a largely overlapping set of loci with partially different phenotype definitions. I would therefore encourage the authors to:

- Clearly articulate, in the Abstract and Discussion, what they view as the main novel biological or methodological contribution beyond Zhao et al. and other prior FC GWAS that used UK Biobank data. For example, did the edge provide more information than ICA in previous studies?
- Be cautious with the use of terms such as "novel loci" or "new genetic factors" when the same regions/genes have already been reported for functional connectivity or closely related traits and instead frame the contribution as a more fine-grained characterisation or contextualization of previously known loci where appropriate.

Clarifying these points would help readers better appreciate what unique insight this study adds on top of the existing literature.

2. What does an “edge” represent biologically?

A central conceptual issue for me is how to interpret an individual edge-level association biologically. In your current framing, edges are treated as 3,321 distinct phenotypes, but it remains unclear what a statistically significant association with a single edge (or a small subset of edges) should be taken to mean in terms of underlying neurobiology. This concern is amplified by your own results showing that the major loci have quite widespread effects across many edges, rather than being highly specific.

It would be very helpful if the authors could expand the Discussion to:

- Explain, conceptually, how they think about an “edge” as a genetic phenotype: is it primarily a proxy for variation in a broader network, for vascular/physiological factors, or for more local circuit-level properties? can we use this edge to explain the
- Clarify to what extent the key conclusions should be interpreted at the edge level versus at the level of subnetworks or global patterns. For instance, do the authors view their main message as “specific edges are associated with gene X” or rather “gene X has a broad, pleiotropic impact on functional connectivity, which can be seen across many edges when looked at in a fine-grained way”?

Some additional text explicitly addressing what an edge-level finding “means” in biological terms would, in my view, substantially improve interpretability for a broad readership.

3. Validity and stability of edge-level genetic architecture

Finally, while you do mention that SNP-heritability correlates with test–retest reliability and that global signal regression improves this relationship, the general issue of fMRI instability and measurement noise remains important for a mass-univariate edge-wise GWAS.

I would therefore encourage the authors to:

- More clearly describe (in the main text, not only in the Supplementary Notes) how reliable the edges are on average (e.g., typical ICC values, how the reliability of edges underlying the top loci compares with the rest of the connectome).
- Explicitly discuss in the Discussion how the known sensitivity of rs-fMRI to motion, arousal, physiological noise and preprocessing choices might affect the inferred genetic architecture, and why the authors believe the key findings are robust despite these limitations.

To be clear, I am not necessarily asking for extensive new analyses; rather, I think a more explicit and nuanced discussion of these points would help position the work more accurately in terms of novelty and robustness.

4. Edge–disease relationships and lack of replication on independent or updated samples

In addition, the manuscript also examines relationships between edge-level measures and various clinical or disease-related phenotypes. As far as I can see, the vast majority of these associations do not survive appropriate multiple-testing correction. This raises further questions about the practical validity and utility of edge-wise phenotypes as intermediate traits for disease, beyond their statistical heritability.

It would be helpful if the authors could:

- Be more transparent and explicit in the main text about how many edge–disease associations remain significant after correction, and temper any strong claims about disease relevance accordingly.
- Discuss whether the lack of robust edge–disease associations is primarily a power issue (given the large multiple-testing burden), or whether it may reflect limited construct validity or stability of edge-wise phenotypes as disease proxies.

Overall, I appreciate the substantial additional work the authors have already done. With a clearer articulation of what is genuinely new relative to Zhao et al., a more explicit discussion of how to interpret edge-level genetic associations in the context of noisy rs-fMRI data, and some consideration of disease relevance and replication on independent or updated samples, I believe the manuscript would be considerably strengthened.

Reviewer #2

(Remarks to the Author)

The authors have accurately responded to my comments.

Dear editor,

We would like to thank you and the reviewers for your time and consideration of this manuscript. Reviewers' comments are reproduced in full below. Our responses are added in blue. References to figures and tables in this document are labeled with R: Figures R1-R8 and Table R1.

Changes to the text are marked in blue in the main text and supplementary information. Briefly, **Supplementary Note 5**, and **Supplementary Tables 5 and 6** were added. **Supplementary Methods, Supplementary Notes 3 and 6, Figure 1 and Figure 2** were updated. Supplementary display items were renumbered to follow their presentation order in the main text.

Sincerely,
On behalf of the authors,
Bernardo Maciel & Martijn van den Heuvel

REVIEWER COMMENTS

Reviewer #1 (Remarks to the Author):

This study presents a large-scale genome-wide association study (GWAS) of pairwise functional brain connectivity (FC) across 3,321 interregional connections in ~28,000 participants from the UK Biobank. The authors identify several loci and genes (e.g., EphA3, PAX8, APOE, SLC39A12) with pleiotropic effects on FC, and demonstrate genetic overlap with neuropsychiatric, cardiovascular, and cognitive traits.

While the study is ambitious and methodologically rigorous, I have concerns regarding its interpretability, and application.

Major:

1. Previous studies have already investigated the genetic architecture of functional brain connectivity (e.g., Zhao et al., 2022). In contrast to these approaches, which integrate multiscale network-level traits, this study performs mass-univariate GWAS across over 3,000 pairwise edges. While this design is fine-grained, it also introduces an enormous multiple testing burden, likely diluting statistical power and inflating false negatives.

The authors should consider comparing their findings directly with those from Zhao et al., and/or apply multivariate GWAS techniques (e.g., MOSTest, Sha et al. Science Advance, 2023) to validate their key results. Additionally, the generally low heritability values across traits deserve further explanation.

Answer:

Thank you for your compliments on the rigour and ambition of our study.

We agree that applying a fine-grained parcellation across 3,321 pairwise edges entails a substantial multiple-testing burden, which can reduce statistical power and increase false negatives. To assess the impact of this choice on the results reported in this manuscript, we have compared our results with Zhao et al., as suggested above. We compiled all genome-wide significant (GWS; $p < 5 \times 10^{-8}$) associations with edge-wise functional connectivity reported by Zhao et al., totaling 232 unique GWS loci. These were compared against the 887 GWS loci identified in our study. We defined overlap

between studies as loci with boundaries overlapping within a 250 kb window, allowing for slight variations in LD structure.

This comparison revealed substantial concordance between our findings and those of Zhao et al. Out of all GWS loci in our study, 33% were also reported by Zhao et al. Among the four loci reaching study-wide significance (SWS; $p < 1.5 \times 10^{-11}$) in our study, three were also SWS in Zhao et al., and the fourth was GWS. Conversely, 71% of Zhao et al. GWS loci were replicated at GWS in our dataset. In addition, we detect an additional loci 594 at GWS level which were not previously reported for edge-wise connectivity by Zhao et al. **Figure R1** (below) illustrates the overlap of GWS loci across the two studies.

Taken together, these results indicate that despite the increased multiple-testing burden, our fine-grained approach reproduces the majority of previously reported loci while also finding and highlighting new associations. We chose to focus our main analyses and interpretations on the most statistically robust signals, while emphasizing that the full set of GWAS results can support future hypothesis-driven analyses at appropriate thresholds.

The following changes were made to the manuscript to address this comment. A new supplementary note summarising the results above was added (**Supplementary Note 5**) together with **Figure R1**. The following text was added to the main text in the **Results** section (line 121):

*“To evaluate the impact of different phenotypic definitions, we compared our results with GWAS on functional connectivity in literature. Seventy-seven percent of genome-wide significant (GWS; i.e., $p < 5 \times 10^{-8}$) findings in a study using Independent Component Analysis (ICA)-based edge definition also had GWS associations with our phenotypes. (**Suppl. Note 5**)”.*

The following text was adapted in the **Discussion** section (line 292):

“First, a mass-univariate approach entails a substantial multiple-testing burden, which reduces power and may increase the false negative rate.”

Figure R1. Comparison of GWS loci with Zhao et al. Miami plot of all lead SNPs reported in the present manuscript (top) and in Zhao et al. (bottom). Only the lead SNPs are shown. In case a SNP was reported for two different traits, the lowest p-value is represented.

2. The proposed “effect extent” score is an interesting concept for quantifying the breadth of gene influence across traits, but it is currently presented without sufficient statistical justification. The metric does not appear to correct for phenotype correlations, or the inherent arbitrariness of the p-value thresholding used to define “significant” edges. Without replication or statistical modeling of trait dependency, this measure should be interpreted with greater caution.

Answer:

Thank you for your remark. We agree with this point and have revised our approach to improve the interpretability of our effect extent metric. We have also interpreted this measure with greater caution by removing mentions of untested comparisons between effect extents, which would require the development of a null model.

Furthermore, we now present the effect extent as the ratio of significant edges relative to the full connectome. This formulation provides a more intuitive interpretation as the proportion of the connectome expected to be associated with a given genetic variant. Regarding the arbitrariness of p-value thresholding, we chose to report the effect extent based on FDR-significant associations at a 5% false discovery rate because we believe that this approach provides the best balance between interpretability and sensitivity in our context. FDR controls the expected proportion of false positives among discoveries while maintaining power across correlated phenotypes, which we find appropriate here given the trait dependencies, as mentioned above. In contrast, Bonferroni correction assumes independence, which may hide potentially meaningful variation we aim to quantify.

In the previous version of the manuscript, we have also tested whether the average effect extent of disorder-associated genes was higher than expected for a random set of genes. To this end, a null model was applied but this was not clear in the main text. We have now moved the full description of the null models to a dedicated section in the **Supplementary Methods**. Briefly, we calculate the effect extent for each gene of the connectome by applying the definition above to genome-wide gene association studies (GWAS) performed with MAGMA. Then, for each disorder, we calculate the average effect extent of the gene-set composed by all its GWS genes. Letting N be the size of the gene-set, we build a null model by sampling 10,000 gene-sets, each with N random genes. To avoid bias introduced by linkage disequilibrium (LD) in the sampling procedure, the gene-sets are sampled from a multivariate normal distribution with a root matrix equal to the gene-gene correlation matrix observed in our discovery sample (.raw file in MAGMA). To further improve on this sampling procedure, we have also created a null distribution by sampling only from highly brain-expressed genes (6,401 genes), as defined by SynGO (Koopmans et al. 2019; PMID: 31171447).

The **Results** were edited as follows to remove mentions of untested comparisons between absolute effect extent values and integrate the above comments (line 140):

Given FC-associated genes showed a widespread connectome impact, we investigated whether genes implicated in neuropsychiatric disorders show greater effect extent in FC than expected beyond highly brain-expressed genes (Suppl. Methods; Suppl. Note 6). We calculated the expected effect extent of genes previously associated with seven psychiatric conditions. A significantly larger extent was found for genes linked to anorexia nervosa ($p = 0.03$), Alzheimer’s disease (AD; $p < 1 \times 10^{-4}$), and schizophrenia ($p < 1 \times 10^{-4}$). The gene-sets for AD and schizophrenia remained significant after correcting for multiple testing ($\alpha = 0.05/7$ disorders tested) (Suppl. Fig. 7; Suppl. Table 6).

Below follows the relevant revised text in the **Methods** section (line 409):

A total of 1490 locus-edge associations (864 unique lead SNPs) were discovered at GWS level and further assessed across all brain connections to estimate the effect extent of these loci throughout the brain. For each of these 864 loci, the p-values for all SNPs were combined using a SNP-wise mean model in MAGMA v1.10, resulting in one p-value per locus for each edge. Benjamini-Hochberg FDR correction for the number of traits (3,321) was applied across edges to these p-values to quantify with how many traits each locus was expected to be associated. The ratio between the FDR-significant and total number of tests is referred to as the effect extent of a locus.

Supplementary Methods were updated according to the response above. **Figure 2**, **Supplementary Figure 2**, **Supplementary Note 6** were also updated with the results of the revised metric which carried no substantial changes in the interpretation of the results.

3. The brain is inherently organized at multiple scales (e.g., modules, networks, gradients). This study focuses exclusively on edge-level analysis, without integrating these higher-order structures. As a result, it is difficult to interpret the biological relevance of the findings within the broader context of brain functional architecture.

Answer:

The aggregation of edges into higher-order structures was conducted post hoc to help with the interpretation of our findings and to assess enrichment of SNP-heritability within specific resting-state networks (RSNs). Our group has previously examined the multiscale organization of the brain at both genetic (e.g., Tissink et al., 2023; PMID: 36882310) and phenotypic levels (e.g., Scholtens et al., 2014; PMID: 25186762). Modules and networks are often defined as collections of nodes. In contrast, the present work uses an edge-based approach to investigate genetic contributions to connections between nodes. This allows us to detect genetic influences that may be specific to inter-regional connectivity and not apparent when signals are averaged within larger structures. To further expand on the comment above, we compared the polygenic signal of RSNs and edge GWAS.

To compare the genetics of individual edges with the broader network structures they connect, we used LDSC to estimate genetic correlations between each edge-GWAS and GWAS of RSNs FC from Tissink et al. (2023). The *p*-values of the genetic correlation estimates were FDR-corrected, and the significance threshold was set at 0.05/7 to account for the seven RSNs tested.

We identified 11 significant genetic correlations with the somatomotor network, 20 with the default mode network (DMN), and 4 with the limbic network. All significant correlations with the somatomotor network involved edges within the network. Interestingly, the strongest correlations were between the bilateral pre-, para- and postcentral gyri, suggesting that global somatomotor signal mostly captures primary sensorimotor function. For the DMN, 6 of the 20 significant correlations were with edges belonging to the DMN, while the remaining 14 involved edges between networks. In the limbic network, 2 of the 4 significant correlations were with edges within the network, and the other 2 with edges between networks. Additionally, for the ventral attention network, a number of edges showed subthreshold correlations that suggest a similar pattern, indicating that some network-specific correlations may not reach significance due to limited power or lower heritability of individual edges (see **Figure R2** for a summary of the results). All significant or suggestive within-network associations were positively correlated with the respective RSN, as expected. Overall, these findings indicate there is convergence and unique information of edge-level and network-level genetic architecture, indicating that these two approaches provide complementary information.

The following changes were made to integrate this analysis in the manuscript. The following sentence was added to the **Results** section, summarising the results above (line 125):

Genetic correlation analysis using RSN-level GWAS suggests that edges within a given network are generally more genetically similar to the overall signal of that network, indicating convergence of genetic architecture at different scales of FC analysis.

The text above and **Figure R2** were added to **Supplementary Note 5** in section *Comparison with RSNs*.

Figure R2. Similarity of edge and RSN genetic architecture. LDSC genetic correlations between GWAS were performed on FC of RSN and in each of 3,321 edges included in this study. All q -values presented in the figure correspond to the Benjamini-Hochberg FDR-corrected p -values of the LDSC genetic correlation estimate.

4. *The discussion of pleiotropic genes is largely descriptive and lacks quantitative metrics to assess the degree of pleiotropy across edges. The authors could improve this by applying phenotypic clustering, genetic correlation matrices, or network-based enrichment analyses to identify modules or subnetworks consistently influenced by shared genetic variation.*

Answer:

Thank you for your comment. We agree that quantifying the number of different networks consistently influenced by the same genetic variation is a natural description of pleiotropy within our study of a network of traits.

To add to the quantification of pleiotropy through the effect extent metric discussed above, we have calculated genetic correlations between each edge and constructed a 3321 x 3321 cross-edge genetic correlation matrix, as suggested. After filtering for unbounded estimates (due to low/non-significant heritability of some edges), 983 edges remained (**Figure R3**). Next, we tried several approaches for the quantification of modularity of this matrix. Appropriate modelling of the genetic covariance of these traits (and associated errors) would require a model such as genomic SEM, which is not computationally feasible for the number of input traits. As an alternative, we capped the genetic estimations at -1 and 1 and computed the nearest positive definite matrix using Higham's algorithm. This matrix was then used for factor analysis, but did not converge, likely due to the numerical imprecisions in the estimation of each individual correlation. To surpass these technical difficulties, we performed a principal component analysis (PCA) instead. We limited our analysis to quantifying modularity and the number of independent components because this method does not fully model the uncertainty in the estimation of each correlation. Community structure was calculated using Louvain's algorithm. Clustering of the correlation matrix was performed using Ward's method with Euclidean norm as the defining distance.

Out of the 983 components in the matrix, 221, 338 and 424 explained 95, 99 and 99.9% of the variance of the data (**Figure R4**). The optimal structure of the network detected 5 different communities with an associated modularity (Q) value of 0.11, which indicates a low modular structure of the genetic correlational structure. These estimates of low modularity likely reflect errors in the numeric estimation of the correlation matrix.

The edge-edge genetic correlation matrix was correlated with the phenotypic correlation matrices to gauge similarity between phenotypic and genetic variation. Genetic and phenotypic structures correlate strongly ($r = 0.61$, $p < 2.2 \times 10^{-16}$).

Altogether, we concluded that the current estimations of genetic correlations are too noisy to perform network analyses. Increasing sample sizes will provide better estimations of genetic correlations and improve this analysis.

Figure R3. Cross-edge genetic correlation matrix. Each element corresponds to the correlation between two edges. Clustering performed equally on rows and columns using Ward's method.

Figure R4. Scree plot of genetic correlation matrix PCA. In blue are the eigenvalues associated with each component. In green is the cumulative variance of the principal components. PCA performed on the nearest positive definite matrix of the genetic correlation matrix, computed using Higham's algorithm. Graph truncated at principal component 300 of 983.

5. *Although some genes implicated in FC are also associated with major brain disorders, the manuscript stops short of establishing clear mechanistic pathways. Simply observing overlapping loci or tissue enrichment does not sufficiently establish FC as an intermediate phenotype. A more systematic investigation—such as polygenic risk scoring (PRS), LDSC-based genetic correlation, or trait-pathway mediation analysis—would considerably strengthen this claim.*

Answer:

Thank you for your comment. While connections between specific genes and disorders were made in the **Results** and **Discussion** sections, a systematic approach to associate neuropsychiatric genetics and FC had not been explicitly discussed.

We had previously used the effect extent metric to investigate whether genes related with neuropsychiatric disorders are expected to be associated with a higher proportion of the functional connectome than what would be expected by chance, which was found to be significant for schizophrenia and Alzheimer's disease (**Supp. Note 6; Suppl. Fig. 3**). To complement this analysis, we ran LDSC genetic correlations between all edges and the 7 neuropsychiatric traits used (**Figure R5**). Only schizophrenia showed FDR-significant genetic correlations with 21 edges (**Figure R6**). All other traits presented at least one nominally significant correlation, but none survived multiple-testing correction. Similar to the analysis described in response to comment #4, we expect these more significant correlations with increasing sample sizes of the FC GWAS.

The following changes were added to the manuscript. A new subsection *LDSC analysis* and **Figures R5-6** were added in **Suppl. Note 6** summarising the text above. The following sentence was added to the **Results** section (line 186):

Next, we investigated whether FC was genetically correlated with neuropsychiatric diagnoses using LDSC. Out of all edges, 21 showed a significant correlation with schizophrenia (FDR-corrected $\alpha = 0.05$). No other association survived multiple-testing correction (Suppl. Note 6; Suppl. Fig. 11; Suppl. Table 10). These results suggest schizophrenia is associated with several functional edges.

Figure R5. Volcano plots of genetic correlations between edge GWAS and disorder GWAS. Each element is a genetic correlation coloured according to its significance. In yellow is $p_{rg} < 0.05$ (Nominal), in red is $q_{rg} < 0.05$, where q -value is the FDR-corrected p -value.

Figure R6. Significant genetic correlations between schizophrenia and FC edges. On the left and right are depicted the areas involved in the negative and positive genetic correlations with schizophrenia, respectively. Only FDR-significant edges (in red) are depicted. Plot replicated from the respective panel on **Figure R5**.

6. While the authors identify genetic variants associated with FC, they do not test causality or directionality. Incorporating Mendelian Randomization (MR), causal mediation models, or network-based gene prioritization could clarify whether SNPs influence clinical traits via connectivity mechanisms. Without such analyses, the study remains largely descriptive and falls short of positioning FC as a valid biomarker for brain disorders.

Answer:

Thank you for the suggestion above. Causality had not previously been tested in this manuscript. To investigate putative causality we have performed MR analyses, as suggested above. We note that a study with an experimental design is more appropriate to investigate causality. MR only provides suggestive evidence of causality, and proving biomarkers remains far outside the scope of a single study.

Generalised Summary-data-based Mendelian Randomization (GSMR2; Xue et al. 2024, PMID: 38472333) was run on all edge-GWAS which showed a nominally significant genetic correlation with a disorder-GWAS as calculated for comment #5. This method was chosen due to its filtering for pleiotropic variants and weak instruments, and high power even with a low number of instrumental variables, which is the case for most of our edge-GWAS. A minimum of 5 independent GWS (i.e., $p < 5 \times 10^{-8}$) SNPs were considered for each analysis. Reverse analyses were only run if at least one significant locus was identified for the respective edge-GWAS. This accounted for 241 GSMR analyses: 6 for AD, 29 for ANO, 55 for BIP, 31 for MDD, 93 for SCZ and 27 for SUD. Both the forward (SNP \rightarrow Edge \rightarrow Disorder) as well as the reverse causality (SNP \rightarrow Disorder \rightarrow Edge) chains were tested.

No edge-disease pair had enough instruments post-filtering to perform the forward analysis. Twenty nominally significant reverse analyses were found: 1 for AD (positive beta), 9 for BIP (mixed sign), 4 for MDD (negative betas), 6 for SCZ (mixed sign) and 1 for SUD (positive beta). The most significant p -value ($= 5 \times 10^{-4}$) was found for a putative causal negative effect of MDD on the edge connecting the right fusiform gyrus and the right caudal medial frontal lobe (**Figure R7**). No test survived FDR correction across the successfully performed analyses. As the forward analysis could

not be performed due to low locus yield, the results of the reverse analysis were not found to be sufficient to draw a conclusion about causality.

The results above were reproduced in **Supplementary Note 6**. The following sentence was added to the **Results** section to acknowledge the results of this analysis (line 189).

Mendelian randomisation (GSMR) was used to investigate whether there is evidence of a causality link between functional connectivity and neuropsychiatric disorders, but this analysis did not have enough statistical power to draw conclusions about bidirectional causality.

Figure R7. Reverse causality estimation for right fusiform and caudal medial frontal gyri. Each dot represents a SNP used as an instrumental variable and the lines the standard error of the estimation.

Minor

1. In Figure 1c: the authors claim enrichment of heritability in the Ventral Attention Network; however, this does not survive multiple testing correction. The rationale for labeling asterisks differently than in Table 1 should be clarified. In Figure 2a: Consider adding a horizontal line to indicate the significance threshold. In Figures 2b–e: Please indicate the number of significant edges for each panel to enhance comparability.
2. The manuscript applies a single atlas to define FC edges but does not discuss the potential impact of atlas choice on results. A brief comment on robustness across parcellations would be helpful.
3. Ensure gene names are consistently italicized throughout the manuscript (e.g., *EphA3*, *PAX8*, *SLC39A12*), as required by scientific conventions.

Answer:

Thank you for your remarks. These were all implemented. Below follows a description of all changes.

- **Table 1** was updated to denote significance in the same way as **Figure 1b**.
- A line denoting SWS was added to **Figure 2a**.
- The effect extent of each locus was added to **Figures 2b–e** according to the revision performed for comment #2.
- A remark on atlas choice was added to the **Discussion** (cf. Reviewer #2's comment #4): "Third, despite different methods of extraction of functional connectivity having been compared, the impact of choice of atlas or of more dynamic connectivity measures has not been examined in this manuscript." (line 304)

- *EphA3* and *SLC39A12* were italicised in the legend of **Figure 3**.

Reviewer #2 (Remarks to the Author):

This manuscript investigates the genetic architecture of functional brain connectivity using data from the UK Biobank. The authors identified 72 loci associated with pleiotropic effects observed across the brain and involved in functional pathways related to neurodevelopment, cardiovascular, and cognitive phenotypes. Overall, this study has strong potential to greatly improve our understanding of the genetic basis of individual differences in functional brain connectivity. Below are several suggestions for improvement.

Introduction

1. The authors state that previous studies have demonstrated high heritability of functional networks. However, this appears inconsistent with the findings of Elliott et al. (2018, Nature), which did not report strong heritability for functional connectivity measures in the UK Biobank. This discrepancy could be better discussed.

Answer:

Thank you for your appreciation of our study and for raising this point.

We revised the text to remove the phrasing of “high heritability” and now refer more generally to “heritability of functional networks.” (line 37)

2. The manuscript emphasizes "pleiotropic genetic signals across brain connections." Did the authors also investigate non-pleiotropic / connection-specific genetic effects? I am not sure why identifying pleiotropic loci associated with FC will help in understanding brain-related disorders. Could you clarify?

Answer:

Thank you for your question. We recognise that this point was not completely clear in our introduction. We have rephrased to remove the emphasis on pleiotropy and better introduce the concept in our introduction.

The mass-univariate methods applied in this manuscript are agnostic to pleiotropy. Most of the analyses, such as GWAS, edge-wise SNP-heritability calculations, GWAS, and effector gene analyses, are conducted per individual edge. However, functional connections between different areas of the brain have different signal-to-noise ratios, and consequently differential power. Loci associated with a greater number of edges are more likely to be detected, as genetic variants influencing many connections will have an increased probability of being identified for at least one edge. Thus, while we report pleiotropic effects, this is a consequence of the method rather than an explicit aim of the study. Although our current study highlights pleiotropic effects, the framework is equally capable of detecting connection-specific associations (like the association with *SLC39A12*; cf. lines 266-272), and future work with larger sample sizes and different brain phenotypes will be well-suited to explore these more localized effects with similar methods.

More broadly, pleiotropic genetic effects might be of special interest because they may point to core biological pathways that influence large-scale brain network organization. Identifying genes that influence multiple connections simultaneously can provide insights into shared mechanisms underlying global functional connectivity (dys)function.

The following changes were made to clarify how pleiotropy plays a role in the study goal and design. To clarify the importance of pleiotropic effects in brain functioning, we have added the following text to the **Introduction** (line 49):

Identifying genetic variants that influence a large share of the functional connectome and related neurocognitive traits is crucial for understanding the core biological pathways that shape large-scale brain network organization and its disruption in disease.

To elaborate on why mostly pleiotropic genetic effects were detected, we have added the following to the limitations on the **Discussion** (line 293):

(...) study designs such as the one performed here are more attuned to detecting pleiotropic genetic effects, as genetic variants influencing many connections will have an increased probability of being identified for at least one edge. This is of note in our context where different signal-to-noise ratios imply differential power to detect genetic effects on each edge. Consequently, locus yield and SNP-heritability estimates should be interpreted within the context of the current power level and the differential sensitivity to detect genetic effects across the brain.

Methods

3. Global signal regression was not applied. Given the ongoing debate about its effects on functional connectivity measures, the authors should consider including sensitivity analyses using GSR. More broadly, running multiverse analyses (e.g., testing alternative preprocessing strategies) would help assess the robustness of the reported genetic associations (see the conclusion of this paper <https://www.nature.com/articles/s41586-020-2314-9>).

Answer:

Thank you for your comment. We did apply global signal regression (GSR), but had not clearly reported this in the **Methods**. This has now been rectified (line 353).

We agree with your broader point about the importance of testing alternative preprocessing strategies. We have performed a comparison with two different connectivity definitions (ICA-based connectivity and network-based connectivity) to reply to reviewer #1's first and third comments above. To better explore the potential impact of GSR in the genetic signal of FC, we have repeated a representative set of GWAS analysis without GSR. First, we identified the connections most affected by its application (**Figure R8** shows the correlation between connectivity with and without GSR in the UKB sample). From all significantly heritable edges in this study (N=1,083), we selected a representative set: the edge with the minimum and maximum change with versus without GSR, and one edge per decile of change (**Table R1**). We then repeated 11 GWAS for these edges and evaluated the effect of GSR by calculating genetic correlations of the same edge with and without the application of GSR.

Out of the 11 GWAS investigated, 10 showed a lower SNP-heritability estimate when GSR was not applied. This is in line with the analyses we have performed that showed that edges with a higher test-retest reliability have higher SNP-heritability estimates (cf. **Supplementary Note 3**). However, in those traits which retained enough heritability to perform genetic correlations, this is very high ($r_g \sim 1$), indicating that the global genetic signal we detect is similar in both analyses.

The following changes were made to integrate these findings in the manuscript. A new subsection *Global signal regression* was added to Supplementary Note 3. The following sentence was added to the **Results** section (line 105):

SNP-heritability was found to be correlated with edgewise test-retest reliability (Spearman's $\rho = 0.68$), and improved by applying global signal regression, implying that measurement noise affects the power to detect associated genetic variants (Suppl. Note 3).

Table R1. Comparison of SNP-heritability estimates after the application of GSR. Decile: Decile of change of measure with and without GSR (e.g 10% correspond to an edge in the bottom 10% change after GSR, Min: most similar edge, Max: most affected edge). Mean Difference: Mean difference of edge strength across all subjects before and after applying GSR. h^2_{SNP} with no GSR: SNP-heritability value before applying GSR. h^2_{SNP} with GSR: SNP-heritability value after applying GSR. Genetic Correlation: LDSC genetic correlation estimate of GWAS before and after applying GSR. Mean row indicates the mean across all 11 traits.

† LDSC Genetic correlation and heritability are unbounded estimators. Therefore, heritability might be lower than 0 and genetic correlations higher than 1, when power is low. Correlations estimations were capped to 1.00 and heritability at 0.00.

Decile	Corresponding Edge	Mean Difference	Phenotypic Correlation	h^2_{SNP} with GSR	h^2_{SNP} with no GSR	Genetic Correlation
Min	ctx-lh-frontalpole ctx-lh-precuneus	-0.025	0.74	0.045	0.012	1.00 [†]
10%	ctx-rh-superiortemporal ctx-lh-entorhinal	0.075	0.77	0.049	0.039	1.00 [†]
20%	ctx-rh-rostralanteriorcingulate ctx-lh-middletemporal	0.11	0.77	0.046	0.00 [†]	NA
30%	ctx-rh-rostralanteriorcingulate ctx-lh-caudalmiddlefrontal	0.15	0.75	0.046	0.022	1.00 [†]
40%	ctx-rh-postcentral Left-Caudate	0.19	0.77	0.058	0.00 [†]	NA
50%	ctx-rh-paracentral ctx-lh-posteriorcingulate	0.22	0.75	0.047	0.00 [†]	NA
60%	ctx-rh-parstriangularis ctx-lh-precentral	0.26	0.75	0.066	0.00 [†]	NA
70%	ctx-rh-rostralmiddlefrontal ctx-lh-isthmuscingulate	0.29	0.79	0.080	0.058	0.97
80%	ctx-rh-inferiortemporal ctx-lh-parsopercularis	0.33	0.68	0.034	0.019	1.00 [†]
90%	ctx-rh-superiorparietal ctx-rh-parsorbitalis	0.39	0.63	0.044	0.017	1.00 [†]
Max	ctx-rh-middletemporal ctx-lh-superiorparietal	0.63	0.62	0.066	0.067	1.00 [†]
Mean	-	0.23	0.67	0.052	0.021	1.00

Figure R8. Global signal regression (GSR) effect comparison. Each element i,j of the matrix provides the Pearson correlation between the strength of the edge connecting area i and j before and after applying GSR. The colours show the resting state networks (RSNs) to which row i and column j belong.

4. The manuscript reports the use of 82 regions from the Desikan-Killiany and ASEG atlases, yet the results refer to the Yeo 7-network parcellation. I am confused here. Could you justify your choice of atlas, particularly in light of more commonly used functionally derived alternatives such as the Schaefer-400 or Yeo-17/7 parcellations?

Answer:

Thank you for your question. While the GWAS analyses were performed on the 82-region Desikan-Killiany (DK) + ASEG atlas, the network-level results reported in the manuscript were obtained by aggregating edges according to the Yeo 7-network definitions, as described in the section *Subnetwork Definition* in **Supplementary Methods**. In this manuscript, the 7 Yeo-Krienen networks were used post hoc for interpretation and visualization, not as the primary unit of genetic analysis. We have clarified this in the main text to avoid confusion.

Our choice of the atlas was based on future compatibility and to keep the complexity of the current study manageable. The DK atlas is an anatomically based atlas, particularly in studies utilizing FreeSurfer, thus allowing for compatibility with future studies using this parcellation. FreeSurfer outputs are also available to researchers using the UK Biobank, which allows for direct integration of functional and structural data. Additionally, by using 82 regions rather than higher-

resolution atlases like Schaefer-400, we reduce the multiple testing burden in the GWAS, while still providing a full parcellation of the brain. The same analysis on the Schaefer-400 parcellation would entail running 79,800 GWAS, which each would be run using 9,377,860 unique markers. This would entail intractable computational costs and require a much stricter significance level for statistical testing (6.2×10^{-13}) and consequent higher false negative rates, as mentioned in reviewer #1's first comment. Altogether, we believe this approach balances genetic statistical power with biological interpretability.

The following changes were made to the **Results** section of the main text to clarify the use of the 7 Yeo-Krienen networks as a post-hoc aggregation methods (line 87):

Subnetwork enrichment analysis was performed to determine if brain connections within specific resting-state networks (RSNs) were under stronger genetic control than those in other or between RSNs. We aggregated the results for the individual edges into RSNs. RSNs were defined by assigning cortical edges to one of the 7 Yeo-Krienen RSNs or subcortical network if both regions each edge was linking belonged to the same RSN (Suppl. Methods). To account for the higher correlation of edges within the same network, a linear model was developed to aggregate the h^2_{SNP} of individual edges and calculate RSN enrichment for h^2_{SNP} (Methods).

A remark on the generalisability of atlas choice was added the **Discussion** to acknowledge this limitation (line 304):

“Third, despite different methods of extraction of functional connectivity having been compared, the impact of choice of atlas or of more dynamic connectivity measures has not been examined in this manuscript.”

5. Replicating or extending the findings in an independent dataset with distinct demographic characteristics (e.g., ABCD) would greatly strengthen the generalizability and robustness of the results. It would be particularly interesting to assess whether key findings, such as the APOE association, replicate across cohorts.

Answer:

Thank you for your comment. We fully agree that replication in an independent cohort, ideally the younger ABCD cohort, would greatly strengthen the generalizability of our findings. However, as of June 2nd, 2025, the NIMH Data Archive is no longer accepting new or renewal data access requests for ABCD Study data (see: *Notice: ABCD Data Access Requests Update*: <https://abcdstudy.org/scientists/data-sharing/>). There are few other existing and accessible cohorts or publicly available summary statistics that have a large enough neuroimaging sample to robustly replicate our results.

Notwithstanding the importance of the generalisability of the results, we implemented a strict in-sample replication strategy in the current version of the manuscript using a hold-out sample of 3,708 individuals. Replication was defined using conservative criteria, including concordant effect direction and Bonferroni correction across all tested locus-edge and gene-edge associations (cf. **Methods**, subsection *Replication*).

The following text was added to the **Discussion** section to acknowledge the need for a generalisability study (line 303):

Replication in a sample with different demographics (particularly of younger age and diverse ethnic and socioeconomic backgrounds) may clarify the effects of genetic variants across the lifetime and associations with relevant clinical outcomes.

6. The authors focus exclusively on static functional connectivity. Could they elaborate on the rationale behind this choice and discuss the potential for future work using complementary metrics, such as dynamic connectivity, network topology, or graph-theoretical features?

Answer:

Thank you for your comment. We agree with the importance of exploring the genetic basis of complementary functional connectivity metrics.

In the current manuscript, we focused on static functional connectivity (sFC) to maximize statistical power and improve interpretability of the genetic findings. Given the relatively short acquisition time (6 minutes, 490 timepoints), using sFC also helped reduce noise and model complexity input to the GWAS. State-related fluctuations might dilute the stable trait-like signal of FC, making it harder to detect reproducible genetic effects.

This work complements existing studies that have examined other network representations, such as graph-theoretical measures (e.g., Bell et al., 2022, PMID: 36056064), network connectivity (e.g., Tissink et al., 2023, PMID: 36882310), and ICA-component connectivity (e.g., Zhao et al., 2022, PMID: 34140357). By performing a GWAS on sFC using a full-brain parcellation, we provide an interpretable atlas of genetics associations of edge-based FC, as mentioned as an answer to your comment #4. Once reliable associations are identified with sFC, dynamic functional connectivity metrics can be layered on to capture more subtle (state-like) genetic effects, such as those related to behavioral flexibility or psychiatric vulnerability.

We added a remark on this topic in the **Discussion** section of the main text (line 307):

Future work investigating other functional connectivity metrics, such as dynamic functional connectivity, could provide further insights into the state-dependent component of functional connectivity and allow the study of how genes interact with environmental factors.

7. It would be helpful to clarify how the edges were defined (the manuscript does not appear to include any graph-theoretical analyses).

Answer:

Thank you for your comment. A new subsection titled *Phenotype reconstruction* has been added to the manuscript in the **Methods** to describe how the edges were defined.

Briefly, after coregistration of T1 and fMRI images, parcellation, filtering, and regression of global signal and covariates, pairwise Pearson correlations were calculated between the filtered time series of each region for each subject, resulting in a symmetric 82×82 functional connectome. A GWAS was then performed on each unique edge of the matrix across all subjects, for a total of 3,321 unique FC edges.

Below follows the relevant **Methods** text (line 360):

FC matrices were constructed by calculating pairwise Pearson's correlations between the average preprocessed time series of each of the 82 parcellated brain areas. Subjects for which average head motion parameters (across space and time), signal-to-noise ratio or discrepancy between T1-weighted and fMRI scan deviated from the median more than five times their median absolute variation were excluded from the analysis. Each individual element of the FC matrix across all subjects was used as a phenotype for a SNP-based

GWAS. Phenotypes were standardised across subjects and residualised for total intracranial volume.

Dear editor,

We would like to thank you and the reviewers for your time and consideration of this manuscript. Reviewers' comments are reproduced in full below. Changes to the text are marked in blue in the main text and supplementary information.

Sincerely,

Bernardo Maciel & Martijn van den Heuvel, on behalf of the authors.

REVIEWER COMMENTS

Reviewer #1 (Remarks to the Author):

Thank you very much for the detailed and thoughtful response, as well as for performing the additional comparison. I appreciate the effort that went into these revisions.

However, I still have a few remaining concerns that I would like the authors to address more explicitly in the manuscript.

1. Novelty relative to Zhao et al.

From your new analysis, a large proportion of the genome-wide significant (GWS) loci reported by Zhao et al. are replicated in your dataset (you report ~70–77% replication, depending on the definition), and one third of your own GWS loci overlap with Zhao et al. within 250 kb. At the same time, many of your strongest loci (including those reaching study-wide significance) map to genes such as PAX8, EphA3, THBS1 and APOE that have already been implicated in functional connectivity or related imaging phenotypes in previous UK Biobank studies.

As a reader, this makes it somewhat difficult to understand what is truly new about the present work beyond (i) adopting a finer-grained edge-wise parcellation and (ii) re-detecting a largely overlapping set of loci with partially different phenotype definitions. I would therefore encourage the authors to:

- Clearly articulate, in the Abstract and Discussion, what they view as the main novel biological or methodological contribution beyond Zhao et al. and other prior FC GWAS that used UK Biobank data. For example, did the edge provide more information than ICA in previous studies?*
- Be cautious with the use of terms such as “novel loci” or “new genetic factors” when the same regions/genes have already been reported for functional connectivity or closely related traits and instead frame the contribution as a more fine-grained characterisation or contextualization of previously known loci where appropriate.*

Clarifying these points would help readers better appreciate what unique insight this study adds on top of the existing literature.

Thank you for your appreciation and thoughtful comments. We appreciate your time and your contribution to our manuscript.

We agree with the need to highlight the novelty of the study. This study focuses the analysis on discovering and interpreting the genetic basis of functional connectivity (FC). Previous studies provided associations with several aspects of functional connectivity—ICA components, resting-state networks, specific circuitry, graph metrics derived from the functional connectome. These approaches uncovered loci associated with FC, but have low spatial resolution, which limits the understanding of which brain regions are associated with different genetic factors. Here, we adopted a fine-grained anatomically-based parcellation to address this limitation. We found that we replicated most of the associations reported in previous studies using different phenotype

definitions and preprocessing methods. However, our approach allowed us to pinpoint which anatomical landmarks are most strongly associated with each of our findings (e.g., *SLC39A12* with the bilateral putamen, *APOE* associated with right Inferior Parietal and left Lateral Occipital).

Next, we built upon having a full parcellation of the brain to analyse the specificity of each association, that is, to quantify how much each significant locus influences the connectome. Rather than treating each locus–trait hit in isolation, we examined the extent of the effect of each locus across all 3,321 connections, which revealed that the top loci tend to have a widespread effect on many edges. This allows us to contextualise the specificity of these associations. For example, *EphA3* and *THBS1* were discussed in the context of language circuitry before (Mekki, 2022; PMID: 34929384), but we find these genes to be associated with 9% and 3% of the brain, suggesting this association spans beyond these circuitry.

In addition to the above, we applied gene mapping and colocalisation techniques to translate our SNP-level findings into biological interpretations. For example, the association with *SLC39A12* only was found by aggregating SNP effects, and the association with *PAX8* was mapped and colocalised in the same variant for all 15 edges. These gene mapping techniques revealed that the genes influencing FC are enriched for association with cognitive, cardiovascular, lipid/metabolic traits. We believe these analyses clarify how genetic factors influence functional connectivity and localise genetic effects within shared biological pathways across brain and systemic traits.

The following changes were made to the manuscript. The novel contribution of this study was further clarified in the **Abstract**, **Introduction** and **Discussion** section:

Common genetic variants explained individual differences in 33% of all 3,321 inter-regional functional pathways, representing connectivity between 82 brain regions. Seventy-two study-wide significant associations were identified, largely reflecting widespread, pleiotropic effects across the connectome, and were mapped to five genes: (...) (lines 20-22)

Our findings show that the genetic component of individual differences in functional brain connectivity is largely shared throughout the brain, highlighting the importance of genetic variation in large-scale brain organisation and its relation with cognitive function and overall health. (lines 26-29)

While previous studies have identified several loci associated with specific functional circuitry, resting state networks, independent functional activity components, and functional graph metrics, these often have low spatial resolution, limiting understanding of the specific regional connections that may be critical for brain health. (lines 48-52)

We directly map genetic effects onto specific areas of the brain and quantify their extent from localised circuit-level associations to widespread pleiotropic influences. Individual edge-wise GWAS were mapped and summarised using gene-based mapping techniques and showed that genetic associations with FC are shared with both physiological and cognitive traits, placing the functional connectome in the intersection between brain and physical health. (lines 262-266)

This level of analysis enabled us to interrogate the role of genetic variation in individual differences in FC at several levels of macroscale connectivity organisation—individual edges, modules, and resting-state networks. (lines 270-272)

Our study pinpoints and contextualises gene–edge associations with the functional connections for which this association is the most significant and highlights that these are likely to be pleiotropic beyond the associations found. (lines 292-295)

Mentions to novel loci were rephrased:

SLC39A12 is a previously unreported gene for FC found to be associated with the connection between the left and right putamen. (lines 168,169)

A new gene-edge association of FC was found between SLC39A12 and connectivity in the bilateral putamen (minimum $p_{\text{gene}} = 2.7 \times 10^{-11}$; Suppl. Table 7). (lines 244,245)

In addition to these four genes, we found a previously unreported association of FC with SLC39A12. (...) To our knowledge, this is the first time SLC39A12 has been linked to brain connectivity; it previously only had been identified with brain volumetric features. (lines 291,292,295,296)

2. What does an “edge” represent biologically?

A central conceptual issue for me is how to interpret an individual edge-level association biologically. In your current framing, edges are treated as 3,321 distinct phenotypes, but it remains unclear what a statistically significant association with a single edge (or a small subset of edges) should be taken to mean in terms of underlying neurobiology. This concern is amplified by your own results showing that the major loci have quite widespread effects across many edges, rather than being highly specific.

It would be very helpful if the authors could expand the Discussion to:

- Explain, conceptually, how they think about an “edge” as a genetic phenotype: is it primarily a proxy for variation in a broader network, for vascular/physiological factors, or for more local circuit-level properties? can we use this edge to explain the*
- Clarify to what extent the key conclusions should be interpreted at the edge level versus at the level of subnetworks or global patterns. For instance, do the authors view their main message as “specific edges are associated with gene X” or rather “gene X has a broad, pleiotropic impact on functional connectivity, which can be seen across many edges when looked at in a fine-grained way”?*

Some additional text explicitly addressing what an edge-level finding “means” in biological terms would, in my view, substantially improve interpretability for a broad readership.

Thank you for highlighting this point. We agree that the text is not completely clear in how individual genetic associations should be interpreted.

An individual edge in this manuscript represents a macroscale functional connectivity measurement which quantifies the statistical coupling of time series between two anatomically defined regions. Biologically, a genetic association with an edge may reflect different aspects of this connectivity: the tendency of two brain areas to have coupled activity, physiological factors, and the role of these areas within larger-scale functional networks. Consequently, a genetic association with a single edge should not be interpreted as a highly localised molecular effect, but rather as an indication that genetic variation modulates how strongly two regions participate in coordinated large-scale brain dynamics. The results of our enrichment analyses support this claim: while functional connectivity shares a genetic basis with cognitive and neurologic phenotypes, we also detect associations with other cardiovascular and metabolic phenotypes.

In the previous version of the manuscript, we analysed the extent to which different individual edges are associated with different genetic loci. On a univariate GWAS-level, we found

and replicated study-wide significant (SWS) associations with 5 genes. Having established a statistically sound association between specific edges and genetic variants, we have implemented an effect extent metric. The goal of this analysis was to assess whether there is signal that is missed due to the strictness of SWS threshold. As mentioned in the previous rebuttal letter, we believe this analysis to be appropriate and warranted because of the statistical dependencies within the connectome: given phenotypes are correlated, we investigated whether each association was expected to be specific or widespread. The results of this metric highlight that our study mostly finds genetic variants with a widespread association throughout the brain, which suggests that these loci would be discovered to be associated with more edges in univariate analyses with increased statistical power.

The following changes were made in the manuscript. The **Discussion** was extended to clarify the nature of the genetic associations with edges:

We adopted an edge-wise anatomically-based parcellation of the brain to investigate the genetics of FC, allowing the discovery of the strongest genetic effects with high anatomical precision. This level of analysis enabled us to interrogate the role of genetic variation in individual differences in FC at several levels of macroscale connectivity organisation—individual edges, modules, and resting-state networks. Each edge in this manuscript represents the functional coupling between two brain regions, which not only reflects local interactions between two brain regions but also their participation in large-scale network and physiological processes. Consistent with this interpretation, we find that the strongest loci were expected to influence many edges rather than a small, highly specific subset, indicating the effects of common genetic variation in FC are widespread and pleiotropic. (lines 268-278)

Specific remarks on how the findings should be interpreted were added to the **Results** section. In the context of pleiotropy:

These results indicate these loci have a widespread effect throughout the brain (see Fig. 2b-e for spatial organisation of associations, Suppl. Fig. 3-6), which indicates that the effect of these loci is unlikely to be specific to the edges in which they were detected. (lines 146-149)

A specific cautionary note was added to the limitations to contextualise the specificity of each findings. The relevant portion of text is reproduced below:

Consequently, locus yield and SNP-heritability estimates should be interpreted within the context of the current power level and the sensitivity to detect genetic effects across the brain. In particular, the specificity of each finding should be interpreted cautiously as SWS loci were shown to be expected to have a widespread effect on the brain. (lines 326-329)

3. Validity and stability of edge-level genetic architecture

Finally, while you do mention that SNP-heritability correlates with test–retest reliability and that global signal regression improves this relationship, the general issue of fMRI instability and measurement noise remains important for a mass-univariate edge-wise GWAS.

I would therefore encourage the authors to:

- More clearly describe (in the main text, not only in the Supplementary Notes) how reliable the edges are on average (e.g., typical ICC values, how the reliability of edges underlying the top loci compares with the rest of the connectome).*
- Explicitly discuss in the Discussion how the known sensitivity of rs-fMRI to motion, arousal, physiological noise and preprocessing choices might affect the inferred genetic architecture, and why the authors believe the key findings are robust despite these limitations.*

To be clear, I am not necessarily asking for extensive new analyses; rather, I think a more explicit and nuanced discussion of these points would help position the work more accurately in terms of novelty and robustness.

Thank you for raising this point and for your suggestions. We agree that contextualising our results with measurement reliability will allow for a better interpretation of the statistical robustness of our genetic associations.

We agree with the observation that resting-state fMRI may be sensitive to motion, arousal, physiological noise, and preprocessing choices. However, these factors would primarily affect statistical power rather than the underlying genetic signal, as shown by the analyses on ICC and global signal regression in **Supplementary Note 3**. The main results of this manuscript and their discussion were based on replicated SWS loci with widespread pleiotropic effects. In addition to that, associations of FC with neurocognitive and behavioural phenotypes were found and the proteins coded by the set of identified genes are enriched for protein-protein interactions, indicating these are biologically connected as a group. Altogether, we believe our results to reflect a robust genetic influence on functional connectivity, rather than non-physiological or state-like signals.

The following changes were made to the manuscript to address this comment. Next, average ICC values were discussed explicitly for all edges and all edges with replicated SWS genetic associations both on the main text and **Supplementary Note 3**. **Supplementary Note Figure 3** was replotted to highlight the higher ICC of edges with SWS associations. The relevant text added to the **Results** section is reproduced below:

To assess how measurement reliability influences our results, test–retest reliability was calculated in the sample of individuals for repeated measurements were available. Edges associated with SWS loci showed higher reliability (median ICC = 0.41, range = [0.20; 0.60]) than the set of all edges (median ICC = 0.19; range = [0.0, 0.60]). Broadly, h^2_{SNP} was correlated with edgewise test–retest reliability (Spearman's $\rho = 0.68$), and improved by applying global signal regression, implying that measurement noise affects the power to detect associated genetic variants differentially (Suppl. Note 3). Consequently, locus yield is likely to increase with increasing sample size. (lines 121-128)

The discussion of robustness of our genetic findings was integrated in the **Discussion** section:

This is of note in our context where different signal-to-noise ratios imply differential power to detect genetic effects on each edge, as shown by the higher test–retest reliability of edges with SWS associations. (lines 324-326)

Resting-state fMRI may be sensitive to motion arousal, physiological noise, and preprocessing choices. However, these would mainly have transient state effects on FC and reduce our power to detect genetic associations. Here, we find a set of biologically connected genes with associations with phenotypes of physical and cognitive domains, highlighting the trait nature of macroscale brain connectivity. (lines 340-344)

4. Edge–disease relationships and lack of replication on independent or updated samples

In addition, the manuscript also examines relationships between edge-level measures and various clinical or disease-related phenotypes. As far as I can see, the vast majority of these associations do not survive appropriate multiple-testing correction. This raises further questions about the practical validity and utility of edge-wise phenotypes as intermediate traits for disease, beyond their statistical heritability.

It would be helpful if the authors could:

- *Be more transparent and explicit in the main text about how many edge–disease associations remain significant after correction, and temper any strong claims about disease relevance accordingly.*

- *Discuss whether the lack of robust edge–disease associations is primarily a power issue (given the large multiple-testing burden), or whether it may reflect limited construct validity or stability of edge-wise phenotypes as disease proxies.*

Overall, I appreciate the substantial additional work the authors have already done. With a clearer articulation of what is genuinely new relative to Zhao et al., a more explicit discussion of how to interpret edge-level genetic associations in the context of noisy rs-fMRI data, and some consideration of disease relevance and replication on independent or updated samples, I believe the manuscript would be considerably strengthened.

We would like to thank you again for your time and appreciation of our manuscript. We believe that your contribution and that of reviewer #2 made our manuscript stronger.

We agree with your point about the need to remark that edge–disease associations should be interpreted with caution and given the current power level of this study and in absence of an independent cohort. Furthermore, future work establishing the potential practical validity of these phenotypes to interrogate neuropsychiatric disorders is needed before establishing a causal link.

The following changes were made to the manuscript. The results of edge–disease associations were rephrased to explicitly mention the significance level of significant and a note on the influence of power in these results was added:

Next, we investigated whether FC was genetically correlated with neuropsychiatric diagnoses using LDSC. Out of all edges, 21 showed a significant correlation with schizophrenia (FDR-corrected $\alpha = 0.05$, no association reached Bonferroni significance). No other association survived multiple-testing correction (Suppl. Note 6; Suppl. Fig. 11; Suppl. Table 10). With increasing FC-GWAS sample sizes and consequent higher power and better h^2_{SNP} estimations, we expect to find a higher number of significant genetic associations of FC with neuropsychiatric diagnoses. (lines 197-203)

A specific remark on need of further evidence to clearly establish a clear link between gene–edge–disease associations was added to the limitations section:

Studies in cohorts beyond the UK Biobank and experimental design setups are crucial to further assessing the validity of the link between the genetic bases of FC and neuropsychiatric disorders as well as its utility as a clinical biomarker. (lines 335-337)

Reviewer #2 (Remarks to the Author):

The authors have accurately responded to my comments.

Thank you for your comments and your contribution to the manuscript.